# Tumor-produced and aging-associated oncometabolite methylmalonic acid promotes cancer-associated fibroblast activation to drive metastatic progression

Zhongchi Li[1,2,11], Vivien Low [1,2,11], Valbona Luga[1,3], Janet Sun[1,3], Ethan Earlie[1,4], Bobak Parang[1,2,3], Kripa Shobana Ganesh[1,5], Sungyun Cho[1,2], Jennifer Endress[1,5], Tanya Schild[1,5,9], Mengying Hu [1,6], David Lyden [1,6], Wenbing Jin[7], Chunjun Guo[7], Noah Dephoure [1,5], Lewis C. Cantley [1,3,10], Ashley M. Laughney[1,4,8] & John Blenis [1,2,5] ✉

The systemic metabolic shifts that occur during aging and the local metabolic alterations of a tumor, its stroma and their communication cooperate to establish a unique tumor microenvironment (TME) fostering cancer progression. Here, we show that methylmalonic acid (MMA), an aging-increased oncometabolite also produced by aggressive cancer cells, activates fibroblasts in the TME, which reciprocally secrete IL-6 loaded extracellular vesicles (EVs) that drive cancer progression, drug resistance and metastasis. The cancer-associated fibroblast (CAF)-released EV cargo is modified as a result of reactive oxygen species (ROS) generation and activation of the canonical and non-canonical TGFβ signaling pathways. EV-associated IL-6 functions as a stroma-tumor messenger, activating the JAK/STAT3 and TGFβ signaling pathways in tumor cells and promoting pro-aggressive behaviors. Our findings define the role of MMA in CAF activation to drive metastatic reprogramming, unveiling potential therapeutic avenues to target MMA at the nexus of aging, the tumor microenvironment and metastasis.

Metastasis underlies mortality in the majority of solid-cancer tumors, including lung cancer, the leading cause of cancer death, and melanoma, in which the 5-year survival rate is <15% in patients with metastatic disease[1]. As a problem of aging, metastasis is the number one cause of death in people 60–79 years old and represents a direct avenue to steer interventions for improving cancer survival and extending overall lifespans[2]. Toward this end, much investigative effort continues to focus on mutational, epigenetic, and metabolomic changes within the cancer cell that abet the metastatic process. In this arena, we discovered that methylmalonic acid (MMA), a byproduct of propionate metabolism, is increased in the serum of elderly people and contributes to acquisition of aggressive properties in tumor cells,

uncovering a systemic cause for the link between old age and negative cancer outcomes. In addition to the age-dependent increase in circulatory MMA, we have also demonstrated that tumor cells dysregulate propionate metabolism in order to increase local MMA accumulation, driving cancer progression in an autocrine manner[3].

Considering that MMA is increased both in the aging body and locally through tumor production, the next question was how these high local concentrations of MMA could function in a paracrine fashion. The influence of the tumor microenvironment (TME) on metastatic progression is inextricable from the equation. Within the heterogeneous and dynamic TME network, the exchange of secreted factors such as hormones, enzymes, growth factors, cytokines and

metabolites all facilitate a cooperative tumorigenic and metastatic process between tumor and stroma[4]. Cancer-associated fibroblasts (CAFs) represent critical players in the formation of a favorable TME for cancer progression. In addition to extracellular matrix (ECM) deposition and remodeling, CAFs secrete cytokines, growth factors, and metabolites that influence the behavior and function of tumor cells as well as other stromal components. The concentrations, combinations and efficacy of these secreted molecules can be specifically regulated by their delivery through extracellular vesicles (EVs), although the mechanisms controlling these parameters are not fully understood[5]. In tumor cells, CAF-secreted messengers influence tumor growth, metastasis and drug resistance through multiple underlying processes, including inhibiting apoptosis pathways, induction of stemness programs, or epithelial-to-mesenchymal transition (EMT)[6,7]. Many of the traits that epithelial-like tumor cells acquire through EMT enhance successful remodeling of their surrounding ECM, support invasion through tissue, and promote intravasation across the endothelial barrier into the bloodstream. This is supported by histopathological studies showing that cells at the invasive front of tumors exhibit an EMT phenotype[8,9].

In the present study, we show that MMA, increased in the TME by aging as well as by tumor production, activates stromal fibroblasts to CAFs and induces a secretory phenotype. In turn, EVs secreted by MMA-induced CAFs, harboring IL-6 and other factors, promote an EMT in tumor cells, fostering the acquisition of aggressive traits including drug resistance and increased metastatic formation.

## Results

### MMA secreted from tumor cells activates fibroblasts in the tumor microenvironment

Aberrations in the enzymes downstream of methylmalonyl-CoA in the propionate metabolism pathway, namely methylmalonyl-CoA mutase (MUT), methylmalonyl-CoA epimerase (MCEE), methylmalonic aciduria type A protein (MMAA), or cob(I)yrinic acid a,c-diamide adenosyltransferase (MMAB) result in pathogenic systemic MMA accumulation in methylmalonic acidemias[10–13], and drive cancer drug resistance and metastasis through increased MMA accumulation in vitro and in vivo[3] (Fig. 1a). We profiled the transcripts of these metabolic enzymes in individual cells obtained from resected human lung cancer primary tumors and metastases, and found that tumor cells with reduced expression of these genes were enriched in mesenchymal subpopulations (Fig. 1a, b). Given this, and our previous findings that metastatic inducers drive MMA production and pro-aggressive effects on tumors through dysregulation of propionate metabolism, we wondered if tumor-produced MMA might also act on other cell types in the TME[3]. Fibroblasts comprise the major component of the TME, and in some solid tumors even outnumber malignant cells[14]. We knocked down MUT in A549 lung carcinoma and A375 melanoma cells to simulate MMA accumulation by altered propionate metabolism during early steps of metastasis, and co-cultured these cells with MRC5 lung and BJ dermal fibroblasts, respectively (Fig. 1c–e). Five days of co-culture markedly increased CAF markers in the fibroblasts, suggesting that tumor-produced MMA is secreted and activates fibroblasts in the stroma (Fig. 1f). Conversely, blocking MMA production in A375 cells by knockdown of PCCA, a component of propionyl-CoA-carboxylase, repressed their ability to induce the activation and infiltration of fibroblasts in the tumor in vivo (Fig. s1a–c). Notably, an RNA-sequencing dataset of 501 whole tumors from patient lung squamous cell carcinomas showed a correlation between low MUT, MCEE, MMAA and MMAB levels (indicating high MMA) and high expression of cancer-associated fibroblast markers ACTA1 (encoding for SMA) and FAP (Fig. s1d), suggesting that human tumors with greater levels of MMA do indeed harbor a great proportion of CAFs.

We have previously demonstrated that MMA in the serum is largely encapsulated in lipid vesicles, allowing for accelerated entry

into cells at much lower concentrations compared to free MMA[15]. When we isolated extracellular lipid vesicles from the conditioned media (CM) of MUT-knocked down tumor cells (EVs^shMUT-A549), we found that they indeed carried more MMA compared to control vesicles (EVs^shGFP-A549), and could induce CAF markers when used to treat fibroblasts (Fig. s1e–f). Depletion of these vesicles from the CM of MUT-knocked down cells abolished its ability to induce CAF markers in fibroblasts, confirming that tumor-produced MMA, like the MMA in the serum of elderly people, is delivered and acts on cells through EVs (Fig. s1g).

Treatment of MRC-5 and BJ fibroblasts with exogenous MMA reproduced the effect of co-culture with or EVs from MUT-knockdown tumor cells on CAF marker expression in a dose-dependent manner (Fig. 1g). The ability of exogenous MMA to induce CAF markers in fibroblasts was similar to that of the CM and lipid vesicles from MUT-knocked down tumor cells, as well as other known CAF inducers, including TGFβ (Fig. s1h). Proliferation was not affected by 1 mM MMA treatment, and mildly decreased under 5 mM of MMA (Fig. s1i). We confirmed that MMA activation of CAFs was not simply due to decreased pH or altered TCA cycle flux, as other acids from the propionate metabolism pathway were unable to reproduce the phenotype (Fig. s1j). Intriguingly, MMA also induced CAF production of matrix metalloproteinases (Fig. 1f), which contribute to the ECM remodeling that promotes intravasation of tumor cells into the bloodstream in early stages of metastasis[16].

### MMA-treated fibroblasts secrete EVs to promote pro-aggressive reprogramming in tumor cells

To determine if the secretome of MMA-activated CAFs might direct tumor cell behavior, we cultured tumor cells with CM from vehicle- or MMA-treated fibroblasts (CM^veh-MRC5/BJ and CM^MMA-MRC5/BJ) and observed a marked increase in markers of EMT (Fig. 2a). In addition, co-injection of A549 tumor cells with MMA-treated MRC5 fibroblasts into mice significantly increased the ability of tumor cells to metastasize, indicating that one or more secreted factors from MMA-activated CAFs promotes a pro-metastatic phenotype in cancer cells (Fig. 2b).

Next, we aimed to identify the components of the CM secreted by MMA-treated fibroblasts that was driving the EMT phenotype in tumor cells. EVs are loaded with signaling molecules and genetic material, and function as essential signaling mediators in the TME[17,18]. Considering that MMA is delivered from tumor cells to fibroblast in EVs, we looked to see whether the fibroblast messengers reciprocally driving EMT in tumor cells were also contained in EVs. From MRC-5 lung and BJ dermal fibroblasts, we isolated EVs from the CM after vehicle or MMA treatment (EVs^veh-MRC5 and EVs^veh-BJ, or EVs^MMA-MRC5 and EVs^MMA-BJ, respectively) (Fig. 2c). We did not observe a significant difference in the number or size of EVs secreted by MMA-treated fibroblasts (EVs^MMA-MRC5/BJ) compared to those secreted by vehicle-treated fibroblasts (EVs^veh-MRC5/BJ) (Fig. S2a, b). Survey of extracellular vesicle marker proteins confirmed the purity of these EVs (Fig. S2c). To determine if the CAF-secreted factor driving EMT in tumor cells was being delivered through these structures, we then added EVs^veh-MRC5/BJ or EVs^MMA-MRC5/BJ to their tissue-matched A549 or A375 tumor cells (Fig. 2c). Upon treatment of tumor cells with EVs^MMA-MRC5, we once again observed an increase in EMT markers (Fig. 2d). In contrast, the supernatant from the CM^MMA-MRC5/BJ after isolation of the EVs lost its ability to induce this effect (Fig. S3a). We noted that when A549 tumor cells treated with isolated EVs^MMA-MRC5 were then cultured in normal media, they converted back to an epithelial phenotype after 5 days, highlighting the plasticity of EMT (Fig. S3b). Importantly, when EVs^MMA-MRC5-treated tumor cells were released from EVs^MMA-MRC5 treatment, but subsequently co-cultured in the presence of untreated fibroblasts, the tumor cells maintained their aggressive phenotype (Fig. S3b). This underscores the importance of a positive feedback loop between the tumor and stroma, wherein fibroblast activation drives tumor cell aggression, which reciprocally

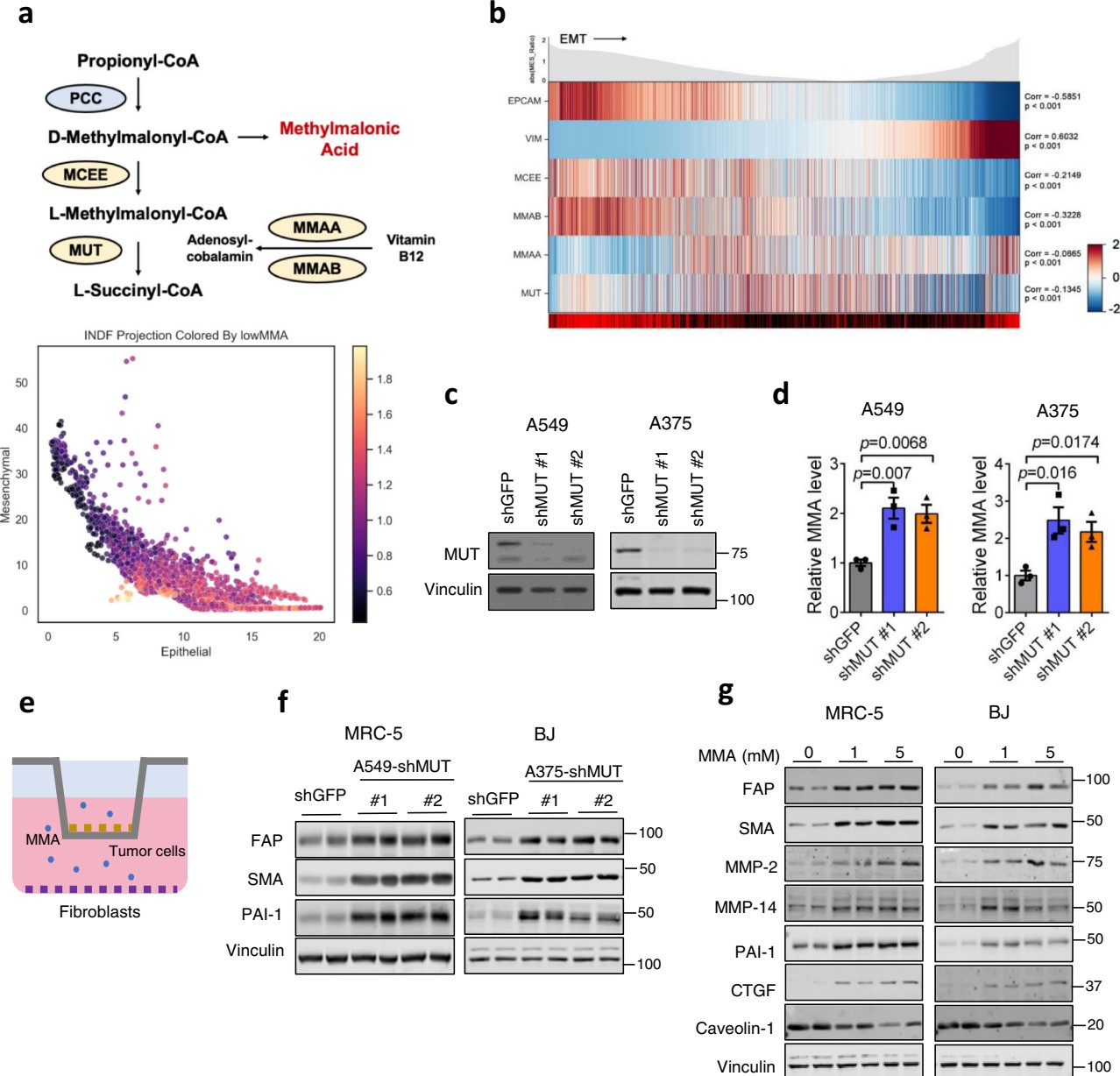

**Fig. 1 | MMA produced by tumor cells promotes a cancer-associated fibroblast phenotype. a** Patient-derived tumor cells ($n = 2537$) projected according to imputed Vimentin expression and imputed EPCAM expression. Each cell is colored by average imputed expression of negative regulators of MMA production (MMAA, MMAB, MCEE, and MUT), highlighted in yellow in the diagram depicting propionate metabolism pathway (above). **b** The z-normalized imputed expression of relevant mesenchymal, epithelial, and MMA marker genes is displayed on the heat map ranked by MES Ratio, defined as the log10 transform of the expression ratio of imputed VIM and EPCAM for each cell. The MES Ratio curve along the top of the heat map shows the absolute value of the MES Ratio across the ranked cells. The color bar along the bottom of the heat map shows the sample tissue source (metastasis: red and primary: black). The gene correlation and associated $p$ values were computed by performing a two-sided spearman test between the normalized (non-imputed) expression of each gene and the MES Ratio. **c** MUT was knocked down in A549 and A375 tumor cells. Immunoblots show the protein level of MUT in cell lysates. **d** MMA levels in the conditioned medium of the tumor cells were measured, normalized to the total cell number ($n = 3$ independent experiments, mean ± SEM, two-sided paired $t$-test). **e** Schematic of the co-culture experiment performed in (**f**). Tumor cells with sh*GFP* or sh*MUT* knockdown were seeded in a transwell insert, and co-cultured with fibroblasts seeded on the bottom of six-well plates. Fibroblast lysates were collected for immunoblots. **f** Immunoblots measuring CAF markers in MRC-5 fibroblasts co-cultured for 4 days with sh*MUT*-knocked down A549 tumor cells and BJ fibroblasts co-cultured for 4 days with sh*MUT*-knocked down A375 tumor cells. **g** Immunoblots measuring CAF markers in MRC-5 and BJ cell lysates treated with 1 mM or 5 mM of MMA for 5 days.

drives more fibroblast activation, ultimately leading to metastatic progression. A375 and A549 tumor cells treated with EVs$^{MMA-MRC5/BJ}$ also exhibited increased resistance to chemotherapeutic and targeted therapy drugs, and displayed increased colony formation in soft agar compared to tumor cells treated with EVs$^{veh-MRC5/BJ}$ (Fig. 2e, f). In addition, tumor cells treated with EVs$^{MMA-MRC5/BJ}$ exhibited increased invasion and migration ability in transwell assays, and formed more

metastases following a subcutaneous primary tumor implantation in vivo (Fig. 2g, h). Intriguingly, despite having significantly higher metastases formation, tumor cells treated with EVs$^{MMA-MRC5}$ did not form significantly larger primary tumors (Fig. 2h). This indicates that the EVs isolated from MMA-activated fibroblasts specifically drive an aggressive, metastatic phenotype in tumor cells, rather than increased cell proliferation.

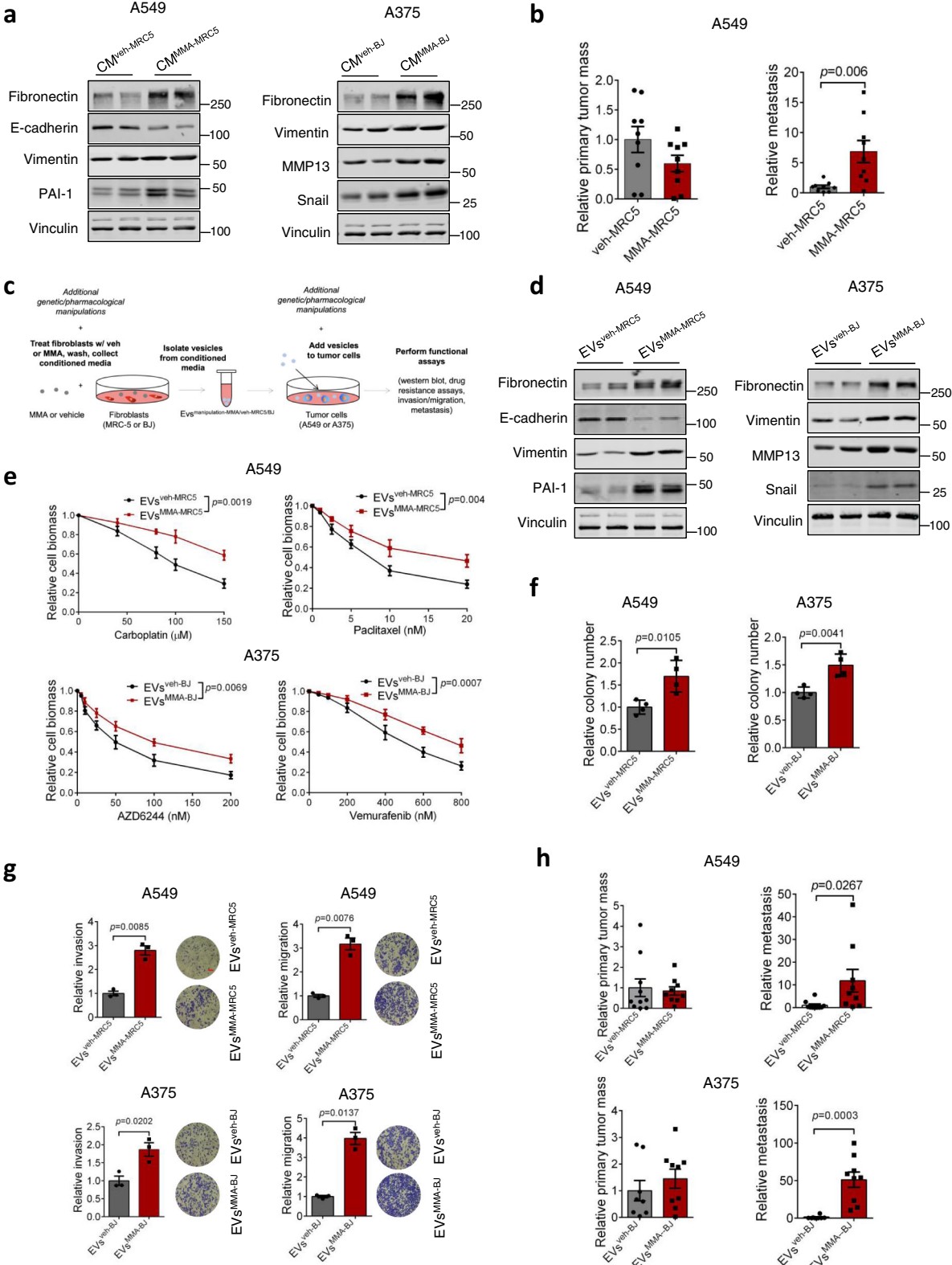

## IL-6 in fibroblast-secreted EVs mediates tumor cell metastatic signaling

As we did not see a change in the number and size of EVs induced by MMA treatment, we speculated that the potent tumor cell response observed after MMA treatment could be due to differentially loaded EV cargo. To identify the active factor in EVs from MMA-treated fibroblasts driving metastatic progression, we performed proteomic analysis on EVs[veh-MRC5] and EVs[MMA-MRC5]. One of the most significantly

upregulated secreted proteins in EVs[MMA-MRC5] compared to EVs[veh-MRC5] was IL-6, a pro-inflammatory cytokine that has been implicated in promoting EMT and metastasis (Fig. 3a, Fig. s4a)[19–21]. We also observed that genes driving IL-6/JAK/STAT3 pathway activity were enriched in more mesenchymal cells characterized by downregulation of key genes restricting MMA production from human lung cancer tumor and metastasis tissue samples (Fig. S4b). Indeed, both IL-6/JAK/STAT3-signaling, measured by JAK2 and STAT3 phosphorylation, and TGFβ

**Fig. 2 | EVs secreted by MMA-treated fibroblasts increase tumor aggressiveness. a** Immunoblots of A549 and A375 tumor cells after 5-day treatment by conditioned media from vehicle or MMA-treated MRC-5 (for A549) or BJ (for A375) fibroblasts. **b** A mixture of vehicle- or MMA-treated MRC5 with A549 cells were injected subcutaneously. The primary tumor and metastasis formation was measured after 6 weeks (n = 9 mice, mean ± SEM, two-sided unpaired t-test). **c** Experimental scheme. **d–h** Pro-aggressive properties in A549 and A375 tumor cells treated with EVs$^{veh\text{-}MRC5/BJ}$ or EVs$^{MMA\text{-}MRC5/BJ}$ from MRC-5 (for A549) or BJ fibroblasts (for A375), evaluated by immunoblots measuring EMT marker expression (**d**), drug resistance assays using carboplatin and paclitaxel for A549 cells, and

vemurafenib and AZD6244 for A375 cells (**e**; n = 3 independent experiments, mean ± SEM, two-way ANOVA), colony formation assays for 3 weeks (**f**; n = 4 independent experiments, mean ± SEM, two-sided paired t-test), transwell invasion and migration assays (**g**; red scale bar indicates 100μM, n = 3 independent experiments, mean ± SEM, two-sided paired t-test) and measurement of primary tumor and metastasis formation 5 weeks after subcutaneous injection into mice (**h**; n = 10 mice for A549 EVs$^{veh\text{-}MRC5}$ group, n = 9 mice for A549 EVs$^{MMA\text{-}MRC5}$ group, n = 8 mice for A375 EVs$^{veh\text{-}BJ}$ group, n = 9 mice for A375 EVs$^{MMA\text{-}BJ}$ group, mean ± SEM, two-sided unpaired t-test).

signaling, measured by phosphorylation of SMAD proteins, were activated in A549 cells upon treatment with EVs$^{MMA\text{-}MRC5}$ (Fig. 3b). Notably, while EVs$^{MMA\text{-}MRC5}$ increased Y705 phosphorylation of STAT3, which is the main regulator of cytokine-induced JAK/STAT3 signaling, it did not affect phosphorylation at S727 (Fig. 3b), suggesting a specificity in EVs$^{MMA\text{-}MRC5}$-mediated downstream signaling. To determine the necessity of these signaling cascades for the ability of EVs$^{MMA\text{-}MRC5}$ to drive EMT, we blocked their activation in A549 tumor cells using the TGFβR or STAT-3 inhibitors, SB431542 and cryptotanshinone, respectively (Fig. 3c). Inhibition of these pathways effectively blocked EMT induction by EVs$^{MMA\text{-}MRC5}$ in A549 tumor cells, re-sensitized cells to drug treatment, and suppressed the increase in invasion and migration (Fig. 3d–f). Similarly, knockdown of *IL6R* in tumor cells suppressed both IL-6/JAK/STAT3 and TGFβ signaling, and suppressed EVs$^{MMA\text{-}MRC5}$-induced EMT marker expression, drug resistance, and invasion and migration, suggesting that IL-6R activation functions upstream of TGFβ pathway signaling in this context (Fig. 3g–j, Fig. S4c). In addition, treating A549 lung tumor cells with tocilizumab, an IL-6R antibody and inhibitor, replicated the effect of *IL6R* knockdown, effectively blocking IL-6/JAK/STAT3 and TGFβ signaling and suppressing the induction of EMT and drug resistance by EVs$^{MMA\text{-}MRC5}$ (Fig. S4d–f). Finally, we knocked down *IL6* in MRC-5 fibroblasts before treating them with MMA and isolated their secreted EVs. While IL-6 knockdown in fibroblasts did not have any effect on the ability of MMA to induce CAF marker expression in fibroblasts, it effectively suppressed the ability of EVs$^{MMA\text{-}MRC5}$ to induce IL-6/JAK/STAT3 and TGFβ signaling in tumor cells, and was sufficient to abolish the EMT-inducing effect of EVs$^{MMA\text{-}MRC5}$ and their ability to boost drug resistance, invasion and migration (Fig. s5).

## MMA activates fibroblasts through ROS activated NF-κB and TGFβ signaling

Next, we set out to characterize the mechanism by which MMA treatment of fibroblasts led to activation of the CAF phenotype and IL-6 loading into and secretion from EVs. We performed RNA-seq on MRC-5 fibroblasts treated with vehicle or MMA, and a pathway enrichment analysis of the RNA-seq data showed an upregulation of genes in the NF-κB and TGFβ signaling pathways in MMA-treated fibroblasts (Fig. 4a). Crosstalk between these two pathways has been described previously, wherein TGFβ signaling leads to the sequential phosphorylation of TAK1, IKK, and NF-κB (Fig. 4b)[22]. We confirmed that these pathways are activated in MRC-5 fibroblasts upon MMA treatment, or treatment by EVs derived from MMA-producing tumor cells (EVs$^{shMUT\text{-}A549}$) (Figs. 4c, s6a). Using time course analysis, we noted that p65 phosphorylation occurred later than SMAD3 and TAK1 phosphorylation (Fig. 4c). Pharmacological inhibition of TGFβR using SB43152, but not of TAK1 and IKK using Takinib and IKK16, respectively, effectively suppressed the induction of CAF markers by MMA, suggesting that the MMA-induced CAF phenotype is largely regulated by TGFβ separately from NF-κB signaling (Fig. 4d, e). Similarly, genetic knockdown of *TGFBR1*, but not *CHUK1* (encoding for IKK1), negated the ability of MMA to induce CAF markers (Fig. s6b, c). Interestingly, pharmacological inhibition of TGFβR, TAK1 and IKK were all individually able to reduce IL-6 loading into EVs$^{MMA\text{-}MRC5}$, indicating that

MMA-induced IL-6 secretion through EVs is mediated by NF-κB downstream of TGFβ-TAK1-IKK activation, and we saw the same effect with genetic knockdown of *TGBR1* and *CHUK1* (Figs. 4f, s6d, e). In line with this and our earlier findings demonstrating the necessity of IL-6, all three inhibitors abrogated the ability of EVs$^{MMA\text{-}MRC5}$ to induce IL-6/JAK/STAT3 and TGFβ signaling in A549 tumor cells, along with EMT (Fig. 4g, h, Fig. s6f, g). In addition, all three inhibitors were able to suppress the ability of EVs$^{MMA\text{-}MRC5}$ to increase drug resistance in tumor cells, although this effect was small using SB43152 or TAKinib (Fig. 4i).

Notably, IKK inhibition had a greater effect than both TGFβR inhibition or TAK1 inhibition in reducing IL-6 loading into EVs$^{MMA\text{-}MRC5}$, which also corresponded with a greater effect in suppressing the potency of EVs$^{MMA\text{-}MRC5}$ for promoting EMT and drug resistance in A549 tumor cells (Fig. 4f–i). This suggested that the NF-κB activation downstream of TGFβR signaling was supplemented by a certain level of NF-κB activation independent of TGFβR signaling, together producing the full effect of IL-6 loading into EVs$^{MMA\text{-}MRC5}$ and the full potency of EVs$^{MMA\text{-}MRC5}$ to induce EMT and increase drug resistance in tumor cells.

As increased generation of reactive oxygen species (ROS) has been established to trigger both NF-κB and TGFβ signaling[23,24], we conjectured that ROS activation of NF-κB both independently and through TGFβR-TAK1-IKK-NF-κB signaling may be at the apex of the MMA signal that induces the CAF phenotype and function. In addition, pathway enrichment analysis of RNA-seq data showed that the oxidative stress response was upregulated in MMA-treated MRC-5 fibroblasts (Fig. 4a). Indeed, MMA treatment, as well as treatment by EVs from MMA-producing tumor cells (EVs$^{shMUT\text{-}A549}$) increased ROS with peak levels at 6 h, corresponding to the peak in TGFβ and NF-κB signaling, while also increasing malondialdehyde (MDA), a marker of oxidative stress, over several days (Fig. 4c, Fig. 5a, Fig. s7a–c). While ROS induction by MMA was similar to that observed by other ROS inducers, including rotenone, TTFA, and hydrogen peroxide, these other inducers were unable to drive the same level of CAF activation in the fibroblasts (Fig. s7d, e). This suggests that MMA may increase ROS through a specific mechanism, or that MMA activates other processes that work with ROS to induce activation of fibroblasts.

Treatment of these fibroblasts with the antioxidants N-acetylcysteine (NAC) or SkQ1 effectively inhibited MMA induction of NF-κB and TGFβ signaling, along with the MMA-induced increase in CAF markers and increased IL-6 loading into EVs$^{MMA\text{-}MRC5}$ (Fig. 5b–e). When EVs$^{MMA\text{-}MRC5}$ were collected from fibroblasts that were co-treated with antioxidants, they were no longer able to activate IL-6/JAK/STAT3 or TGFβ signaling in A549 tumor cells (Fig. 5f). Consistently, antioxidant treatment of fibroblasts suppressed the ability of EVs$^{MMA\text{-}MRC5}$ to induce the EMT phenotype and increase drug resistance in A549 tumor cells, and reversed the ability of these tumor cells to form metastases in vivo (Fig. 5g–i). Together, our data illustrates a mechanism wherein exposure of fibroblasts to MMA generates ROS and induces oxidative stress, which activates NF-κB and TGFβ signaling. Canonical TGFβ signaling regulates CAF marker expression, while NF-κB signaling, which is activated by ROS both independently of and downstream of TGFβ signaling through TAK1 and IKK, regulates IL-6 association and secretion with vesicles. In tumor cells, IL-6 enriched EVs$^{MMA\text{-}MRC5}$ activates

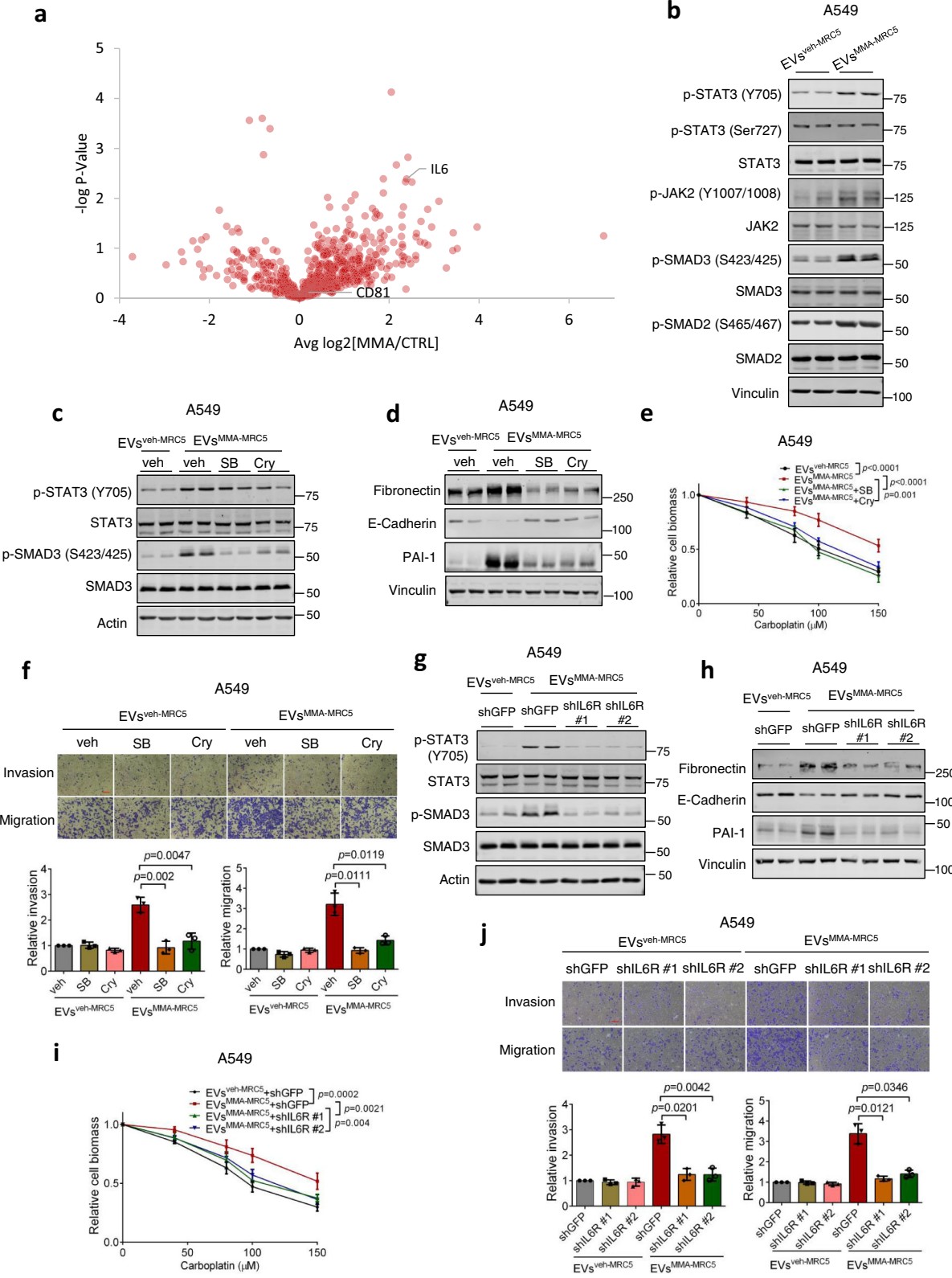

IL-6/JAK/STAT3 and TGFβ signaling, promoting EMT and the acquisition of pro-aggressive traits (Fig. 6).

## Discussion

Here, we depict how a recently identified aging-associated and tumor-produced oncometabolite, MMA, can also function as a tumor cell messenger by acting on the TME to drive metastatic progression. We also characterize the downstream signaling cascades activated by MMA in fibroblasts. By increasing ROS generation, MMA induces a secretory signature in CAFs wherein IL-6 delivered in EVs drives metastatic signaling and progression of epithelial-like (or primary) cancer cells.

While the structure of the signaling cascades activated by MMA are likely to vary according to cell-type, the ability of MMA to induce oxidative stress may be a conserved phenomenon upstream of MMA-mediated functions in other cellular contexts, such as the TGFβ

**Fig. 3 | IL-6 in the EVs of MMA-treated fibroblasts mediates pro-aggressive STAT-3 and TGFβ signaling in tumor cells. a** Volcano plots showing protein level distribution from proteomics analysis comparing the compositions of EVs isolated from vehicle- or MMA-treated MRC-5 fibroblasts ($n = 3$ independent experiments, two-sided paired *t*-test). **b** Immunoblots showing JAK2-STAT3 signaling and TGFβ signaling after 3 h of EV treatment in A549 cells. **c** Immunoblots measuring signal activation in A549 cells pre-treated with vehicle, TGFβR inhibitor SB431542 or JAK inhibitor cryptotanshinone for 30 min, then treated with EVs from MRC-5 fibroblasts for 3 h. **d–f** Pro-aggressive properties of A549 cells treated with EVs[veh-MRC5] or EVs[MMA-MRC5] with or without the TGFβR inhibitor SB431542 or the JAK inhibitor cryptotanshinone for 4 days, evaluated by immunoblots measuring EMT marker expression (**d**), carboplatin resistance assay (**e**; $n = 3$ independent experiments, mean ± SEM, two-way ANOVA), and invasion and migration transwell assays (**f**; red scale bar indicates 100μM, $n = 3$ independent experiments, mean ± SEM, two-sided paired *t*-test). **g** Immunoblots measuring signal activation in A549 cells with sh*GFP* or sh*IL-6R* knockdown (#1 and #2) and treated with EVs[veh-MRC5] or EVs[MMA-MRC5] for 3 h. **h–j** Pro-aggressive properties of A549 cells treated with of EVs[veh-MRC5] or EVs[MMA-MRC5] with sh*GFP* or sh*IL-6R* knockdown for 5 days, evaluated by immunoblots measuring EMT marker expression (**h**), carboplatin resistance assay (**i**; $n = 3$ independent experiments, mean ± SEM, two-way ANOVA), and invasion and migration transwell assays (**j**; red scale bar indicates 100μM, $n = 3$ independent experiments, mean ± SEM, two-sided paired *t*-test).

signaling-dependent increase in SOX4 in MMA-treated tumor cells[15]. More research is warranted to verify this possibility, as well as to elucidate how MMA may increase ROS. A potential mechanism may involve MMA's ability to inhibit succinate dehydrogenase, an essential component of the mitochondrial respiratory chain complex II[25]. Indeed, diseased mitochondria and mitophagy dysfunction has been described in *MUT* deficiency underlying methylmalonic acidemia[26]. Additionally, while previous studies have described a role of ROS in CAF activation, the precise mechanisms by which ROS contributes to this process across various contexts are still unclear[27,28]. For example, ROS induction may be preceded by an upstream event, such as loss of CAV-1, and co-occur with mitochondrial dysfunction and aerobic glycolysis to stimulate CAF activation[27]. In our cellular system, we observed that other known ROS inducers, including rotenone and hydrogen peroxide, were unable to produce CAF marker activation to the same level as MMA. These findings suggest that while ROS is essential for MMA-induced CAF activation, other simultaneous processes, that are potentially also downstream of MMA, are required.

The role of circulatory IL-6 in metastasis and therapy resistance has been previously observed, and is long known to be increased in the serum with age[29,30]. IL-6 was also recently reported to be increased in the serum of methylmalonic acidemia patients[31]. The discovery of a specific mode of IL-6 delivery from stroma to tumor through EVs, however, likely confers a particularly calibrated and potent effect. The proportion of IL-6 released freely or delivered through EVs has been shown to vary widely and depend on the biological systems involved; for example, almost all IL-6 released from monocytes are free, while all IL-6 released from T cells are encapsulated[32]. In addition to providing a concentrated influx of the cytokine when IL-6 is delivered through these lipidic structures, EVs also protect their contents from environmental degradation, and expression of surface proteins may facilitate the targeting of EVs to distinct cell types[32]. Co-delivery of different cytokines encapsulated together is also likely to have different synergistic effects driving distinct phenotypes. A precise characterization of the regulatory mechanisms dictating how IL-6 is loaded into EVs and the proportions of IL-6 encapsulated or embedded in the membranes will require further investigation, and will likely illuminate key pathways for cytokine delivery through EVs in other cellular contexts.

As IL-6 signaling is pro-inflammatory and TGF-β signaling is anti-inflammatory, their associated pathways are often described to function antagonistically. For example, STAT3 can bind Smad3 and disrupt formation of Smad2/Smad3 complexes, hindering the DNA-binding ability of these transcription factors[33]. In contrast, we show that STAT3 signaling in tumor cells treated with EVs[MMA-MRC5] promotes TGF-β signaling in a positive crosstalk interaction to drive EMT. This is supported by previous studies which found that STAT3-Smad3 complex formation and nuclear translocation is required for TGF-β-induced *Snail* promoter activation and EMT induction in *KRAS* mutated Panc-1 cells[34]. The factors that determine the nature of these interactions in different contexts remain to be elucidated and may be defined by the varying strengths of each pathway's activation or by the expression of additional co-factors.

Furthermore, while this was not explored in the current study, our findings support a likely means by which the aging body shapes the TME through MMA, further coloring in the link between age as a risk factor and poorer cancer outcomes. Indeed, it has been shown that older people have increased tissue fibrosis, and the possibility that MMA may play a role in this is certainly intriguing[35]. Beyond the effect of MMA on fibroblasts, the full scope of how MMA functions on other cell components of the TME, such as the immune system, has yet to be uncovered, and represents a huge untapped potential for therapeutic interventions targeting various stages of the tumorigenic process.

## Materials and methods
### Ethics statement
All experiments performed in this study are in compliance with local ethical regulations. All animal experiments were approved by the Institutional Animal Care & Use Committee (IACUC) at the Belfer Research Building Vivarium of Weill Cornell Medicine.

### Analysis of additional validation datasets
The filtered, processed, and annotated single-cell RNA-seq data from an independent lung adenocarcinoma patient cohort[36] was received directly from its authors. This dataset includes 17 patient samples – 5 metastatic, 8 primary tumor, and 4 normal samples. Analysis was limited to annotated tumor cells from 8 primary tumors and 4 tumor metastases; genes not expressed in any of the tumor cells were removed. The one spinal metastasis was excluded because it exhibited stark patient-specific effects within tumor cells that were not reproduced across other samples. This yielded a filtered count matrix containing 2537 tumor cells and 18,947 genes. All analyses were performed using the normalized or normalized-and-imputed count matrices provided by the authors.

### Cell lines
A549 cells (non-small cell lung cancer; CCL-185) were obtained from the American Type Culture Collection (ATCC) and cultured in RPMI 1640 medium (Corning) supplemented with 10% FBS (Sigma-Aldrich) and 1% penicillin-streptomycin (Gibco). A375 human melanoma cells (CRL-1619) were also obtained from ATCC and cultured in high-glucose DMEM (Gibco) supplemented with 10% FBS (Sigma-Aldrich) and 1% penicillin-streptomycin (Gibco). MRC-5 human lung fibroblasts (CCL-171) and BJ foreskin fibroblasts (CRL-2522) were also obtained from ATCC and maintained in EMEM (ATCC) supplemented with 10% FBS (Sigma-Aldrich) and 1% penicillin-streptomycin (Gibco). HEK293T cells were obtained from GenHunter and cultured in high-glucose DMEM (Gibco) supplemented with 10% FBS (Sigma-Aldrich) and 1% penicillin–streptomycin (Gibco). All cell lines were maintained at 37 °C and 5% $CO_2$. All cell lines were routinely tested for mycoplasma and were at all times mycoplasma negative.

### Mice
Female nu/nu athymic mice (Envigo) were purchased at the age of 6–7 weeks, and the experiments were started 7–10 days after the mice were received at the Weill Cornell Medicine Belfer Research Building

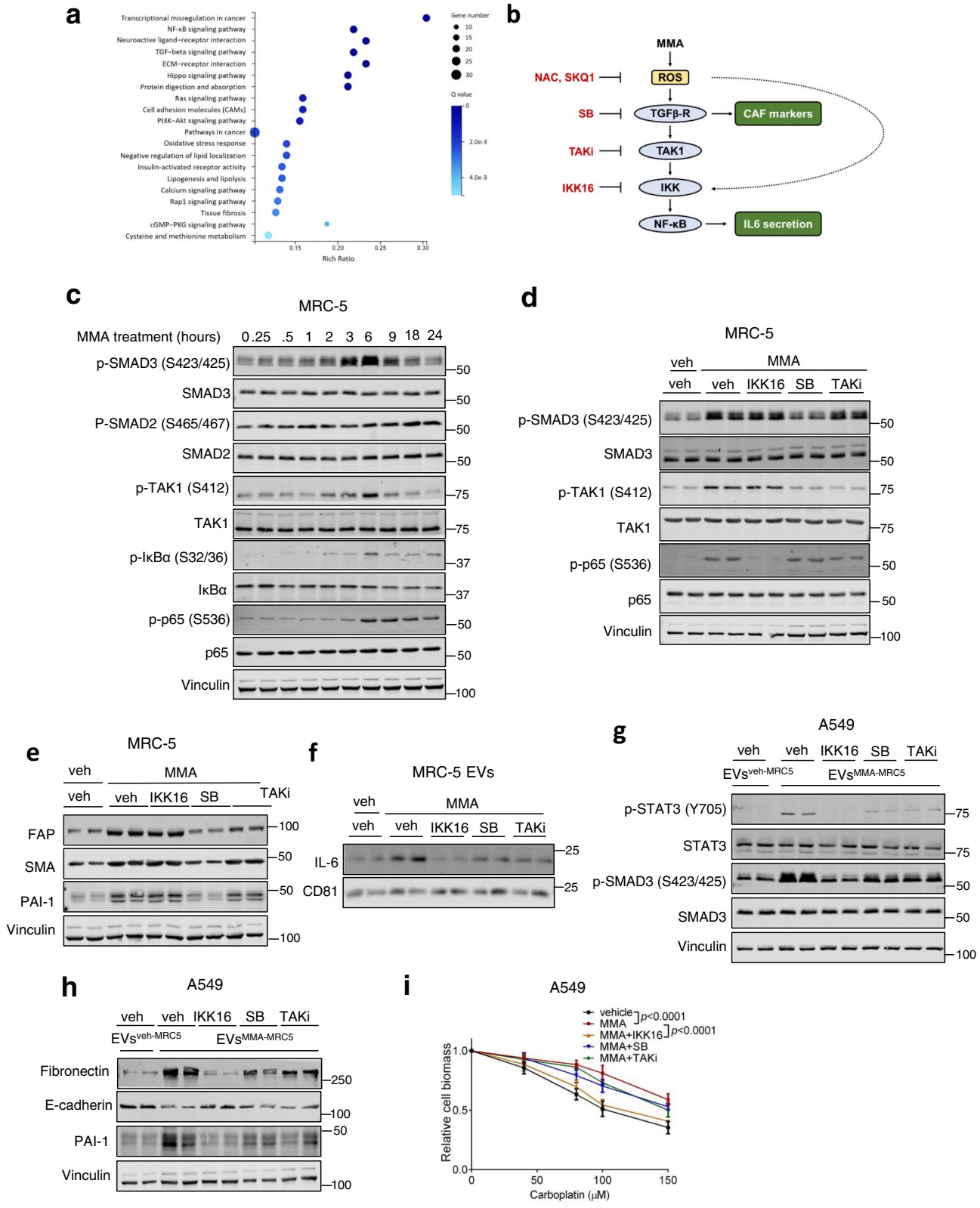

Vivarium. Experimental groups of up to 10 mice were created randomly and mice were group housed (maximum five in a cage) in standard cages with unrestricted acidified water and food (PicoLab Rodent Diet 5053 (Labdiet, Purina) containing 20% protein and 5% fat). Animal husbandry was carried out by the vivarium technical staff in a human xenograft designated area following animal biosafety level-2 procedures. The room was maintained at 21–23 °C with a 12 h light–dark cycle. The mice were maintained in compliance with Weill Cornell Medicine Institutional Animal Care and Use Committee protocols. The tumor size limit on the protocol was 20 mm on the largest dimension or 2.5 cm³ tumor volume or 10% of body weight, whichever was reached first. For mouse studies, no statistical method was used to predetermine sample size, mice were randomly distributed among the treatment groups and no blinding was performed. All mouse studies

**Fig. 4 | MMA promotion of the CAF phenotype and EV-associated IL6 secretion occurs through TGF-β and NF-κB signaling. a** Pathway enrichment analysis on RNA-seq data from MRC-5 cells treated with vehicle or MMA for 5 days. Genes with expression differences greater than two-fold and $p < 0.001$ were counted ($n = 3$ independent experiments, two-sided paired $t$-test). **b** TGFβ signaling leads to downstream NF-κB activation. **c** Immunoblots measuring signal activation over time in MRC-5 fibroblasts treated with 1 mM MMA. **d** Immunoblots measuring signaling activation in MRC-5 fibroblasts treated with MMA alone or in combination with IKK inhibitor IKK16, TGFβR inhibitor SB431542, or TAK1 inhibitor TAKinib for 6 h. **e** Immunoblots of CAF markers in MRC-5 lysates treated with MMA in combination with IKK inhibitor IKK16, TGFβR inhibitor SB431542, or TAK1 inhibitor TAKinib for 5 days. **f** Immunoblots showing IL-6 amount in EVs from MRC-5 fibroblasts treated with vehicle or MMA or MMA alone or in combination with IKK inhibitor IKK16, TGFβR inhibitor SB431542, or TAK1 inhibitor TAKinib. **g** Immunoblots measuring signaling activation in A549 cells treated with EVs from MRC-5 fibroblasts treated with MMA alone or in combination with IKK inhibitor IKK16, TGFβR inhibitor SB431542, or TAK1 inhibitor TAKinib for 3 h. **h, i** Pro-aggressive properties of A549 cells treated with EVs from MRC-5 fibroblasts treated with vehicle, MMA alone, or MMA in combination with IKK inhibitor IKK16, TGFβR inhibitor SB431542, or TAK1 inhibitor TAKinib, evaluated by immunoblots measuring EMT marker expression (**h**) and carboplatin resistance assay (**i**; $n = 3$ independent experiments, mean ± SEM, two-way ANOVA).

have received ethical approval by IACUC at Weill Cornell Medicine under protocol number 2014-0060.

## Immunohistochemistry staining

Immunohistochemistry (IHC) was performed on formalin-fixed, paraffin-embedded tissue. Five-micron paraffin sections were cut. After antigen retrieval with citrate solution, slides were rinsed and blocked with a peroxidase-blocking reagent and incubated with αSMA antibody (ab150301- Abcam, 1:200). Slides were scanned using the Zeiss AxioScan7. Five views per mouse and eight mice per group were used for the quantification. The signal intensity of each view was quantified by using Image J. All the values were normalized to the average signal of sh*GFP* group.

## Conditioned media collection and extracellular vesicle isolation

Fibroblasts were treated with MMA for 5 days. Then, the media was removed, cells were washed twice with PBS, and serum free medium was added. Two days later, the CM was collected. CM was centrifuged at $2000\,g$ for 15 min, then $120,000\,g$ for 20 min. The supernatant was then ultracentrifuged at $100,000\,g$ for 70 min. Then, the supernatant was discarded, and the pellet was washed by resuspension in PBS and re-ultracentrifugation at $100,000\,g$ for 70 min. The total protein amount in EVs was determined using the DC Protein Assay Kit II (BioRad). Particle numbers were determined using the Nanosight NS500.

## Electron microscopy

EVs were processed for EM imaging as previously described[37]. Frozen EVs were thawed and fixed on ice for 5 min in 2% PFA (EMS, 15710). Then, the sample was deposited on Formvar/carbon-coated nickel grids (EMS, FCF400H-NI-SB), fixed for 5 min in 1% glutaraldehyde (EMS, 16120), contrasted for 5 min with 4% uranyl oxalate and finally embedded in 2% methyl cellulose (Sigma, M6385) and uranyl acetate solution (EMS 22400). Images were acquired using a JEOL JEM 1400 transmission electron microscope (JEOL, USA, Inc, Peabody, MA) at 100 keV equipped with a Veleta 2 K x 2 K CCD (EMSIS, GmbH, Muenster, Germany). For each experiment, two grids were prepared for each sample, and 5 random images were acquired from 5 random hexagons in each grid. The largest diameters of all membrane particles were quantitated using ImageJ.

## Co-culture experiments

Cells to be collected for immunoblotting were seeded in 2 ml RPMI media with 10% FBS onto six-well plates. The next day, accompanying co-culture cells were seeded in another 2 ml of RPMI media with 10% FBS onto the top of 0.4 μm cell culture inserts (VWR) placed into the previously seeded six-well plates. The plates were gently shaken four times a day for 3 days. Inserts were discarded and proteins were extracted as described in the *Immunoblots* section.

## Immunoblots

Proteins were isolated directly from intact cells via acid extraction using a 10% TCA solution (10% trichloroacetic acid, 25 mM NH₄OAc,

1 mM EDTA, 10 mM Tris-HCl pH 8.0). Precipitated proteins were harvested and solubilized in a 0.1 M Tris-HCl pH 11 solution containing 3% SDS and boiled for 10–15 min. For EV proteins, samples were extracted using RIPA buffer (40 mM HEPES [pH 7.4], 1 mM EDTA, 120 mM NaCl, 0.5 mM DTT, 10 mM b-glycerophosphate, 1 mM NaF, 1 mM Na3VO4, 0.1% Brij-35, 0.1% deoxycholate, and 0.5% NP-40) supplemented with protease inhibitors (250 mM PMSF, 5 mg/ml pepstatin A, 10 mg/ml leupeptin, and 5 mg/ml aprotinin), incubated at 4 °C for 15 min, then incubated with 4× LDS for 10 min at 70 °C. Protein content was determined with the DC Protein Assay kit II (BioRad), and 30 μg total protein from each sample was run on SDS–PAGE under reducing conditions. The separated proteins were electrophoretically transferred to a nitrocellulose membrane (GE Healthcare), which was blocked in TBS-based Odyssey Blocking buffer (LI-COR). Proteins of interest were probed with specific antibodies (listed as 'target protein' (catalog no. - vendor, dilution factor): FAP (66562 s−Cell Signaling Technology, 1:1000), SMA (ab5694−Abcam, 1:1000), CAV-1(ab2910−Abcam, 1:1000), PAI-1 (612024−BD, 1:1000), PCCA (ab187686−Abcam, 1:2000), Vinculin (V9264−Sigma Aldrich, 1:5000), MMP-2 (4022 S−Cell Signaling Technology 1:1000), MMP-14 (ab51074−Abcam, 1:1000), MMP-13 (ab39012−Abcam, 1:1000), CTGF (ab6992−Abcam, 1:500), E-Cadherin (610181−BD, 1:1000), Fibronectin (ab2413−Abcam, 1:10,000), Vimentin (5741 S−Cell Signaling Technology, 1:2000), Snail (3879 S−Cell Signaling Technology, 1:1000), p-JAK2(Y1007/1008) (3771−Cell Signaling Technology, 1:500), JAK2 (3230−Cell Signaling Technology, 1:1000), p-Stat3 Y705 (ab76315−Abcam, 1:1000), p-Stat3 S727 (9136 S−Cell Signaling Technology, 1:1000), Stat3 (9139 S−Cell Signaling Technology, 1:1000), p-Smad3 S423/425 (ab52903−Abcam, 1:1000), Smad3 (9523 S−Cell Signaling Technology, 1:1000), IL-6 (12153 S−Cell Signaling Technology, 1:1000), CD81 (56039 S−Cell Signaling Technology, 1:1000), CD9 (ab223052−Abcam, 1:500), Flotillin-1 (610820−BD, 1:1000), GM130 (610823−BD, 1:1000), Lamin (4777−Cell Signaling Technology, 1:1000), Calnexin (ab112995−Abcam, 1:1000), β-Actin (4967−Cell Signaling Technology, 1:2000), p-Smad2 S465/467 (3108 S−Cell Signaling Technology, 1:1000), Smad2 (3103 S, Cell Signaling Technology, 1:1000), p65 (8242 S−Cell Signaling Technology, 1:1000), p-p65 S536 (3036 S−Cell Signaling Technology, 1:1000), p-IκBα (Ser32/36) (9246 S−Cell Signaling Technology, 1:1000), IκBα (9242 S−Cell Signaling Technology, 1:1000), TAK1 (5206 S−Cell Signaling Technology, 1:1000), p-TAK1 S412 (9339 S−Cell Signaling Technology, 1:1000). Membranes were incubated with primary antibodies overnight at 4 °C. Membranes were then either incubated with the appropriate horseradish peroxidase-conjugated anti-rabbit (NA934−GE Healthcare, 1:10,000), anti-mouse (NA931−GE Healthcare, 1:10,000) or anti-goat (AP180P−Millipore, 1:10,000) immunoglobulin for 1 h at room temperature, and signals developed using Amersham ECL detection system (GE Healthcare), or they were incubated with the appropriate donkey anti-rabbit Alexa Fluor 488 (A-21206−Thermo Fisher Scientific, 1:10,000) or donkey anti-mouse Alexa Fluor 555 (A31570−Thermo Fisher Scientific, 1:10,000) immunoglobulin for 1 h at room temperature, and signals were developed using the LI-COR Odyssey CLx Imaging System.

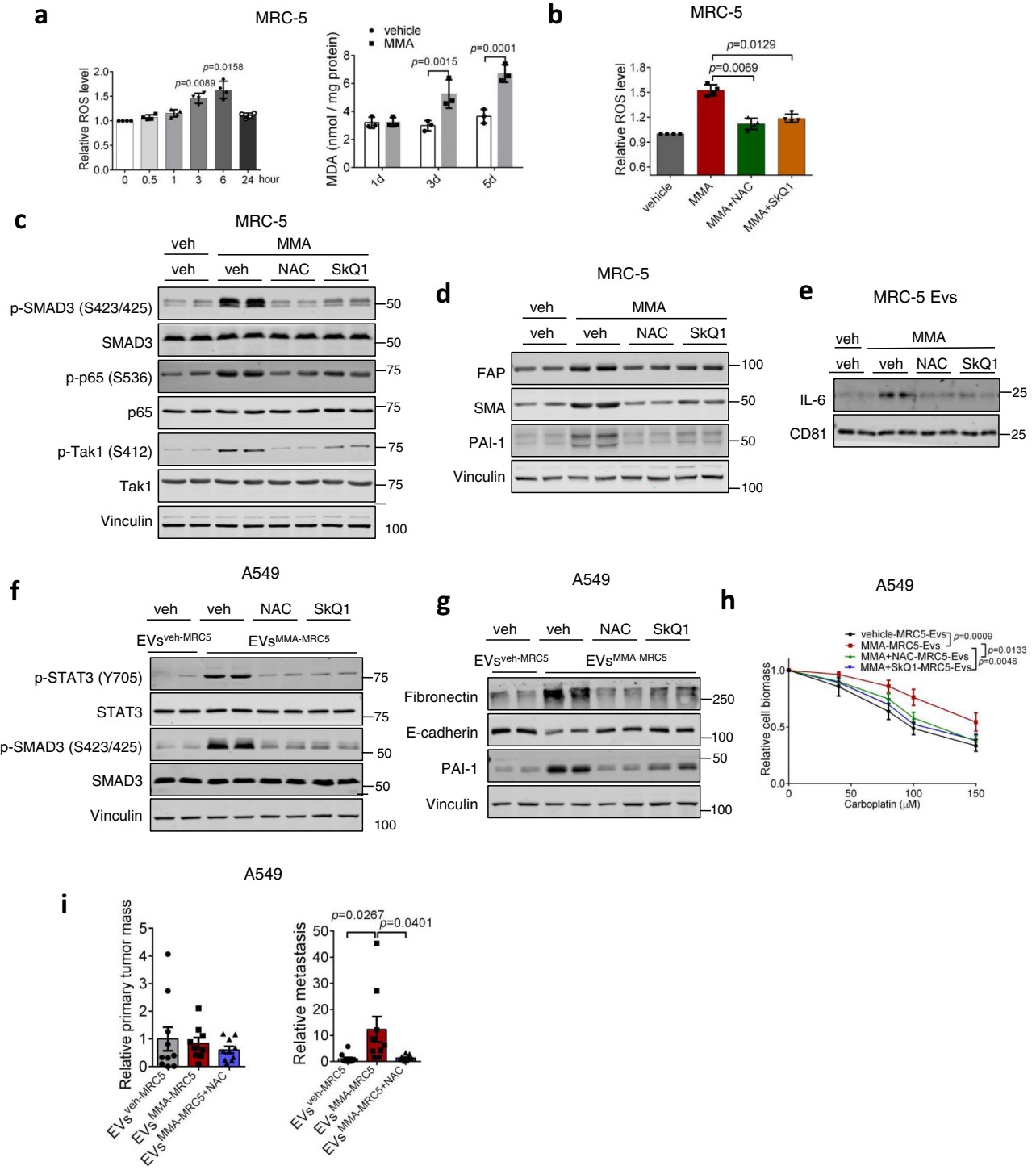

**Fig. 5 | TGF-β and NF-κB mediated activation of fibroblasts and EV-associated IL6 secretion occurs downstream of ROS generation. a** (left) ROS levels and (right) MDA levels in MRC-5 fibroblasts after 1 mM MMA treatment ($n = 4$ independent experiments for ROS measurement, mean ± SEM, one-way ANOVA; $n = 3$ independent experiments for MDA measurement, two-sided paired $t$-test). **b** ROS levels in MRC-5 fibroblasts after MMA treatment alone or in combination with NAC or SkQ1($n = 4$ independent experiments, mean ± SEM, two-sided paired $t$-test). **c**, **d** Immunoblots of MRC-5 fibroblasts treated with MMA alone or in combination with NAC or SkQ1 for 6 h (**c**) or 5 days (**d**). **e** Immunoblots showing IL-6 amount in EVs from MRC-5 fibroblasts after MMA treatment alone or in combination with NAC or SkQ1. **f** Immunoblots measuring signaling activation in A549 cells treated for 3 h

with EVs<sup>veh-MRC5</sup> or EVs<sup>MMA-MRC5</sup>from MRC-5 fibroblasts treated with MMA alone or in combination with NAC or SkQ1. **g**, **h** Pro-aggressive traits of A549 cells treated for 5 days with EVs<sup>veh-MRC5</sup> or EVs<sup>MMA-MRC5</sup> from MRC-5 fibroblasts treated with MMA alone or in combination with NAC or SkQ1, evaluated by immunoblots measuring EMT marker expression (**g**) and carboplatin resistance assay (**h**); $n = 3$ independent experiments, mean ± SEM, two-way ANOVA. **i** Primary tumor and metastasis formation in mice 4 weeks after subcutaneous injection of A549 cells treated with EVs from MRC-5 fibroblasts treated with vehicle, MMA, or MMA and NAC ($n = 10$ mice for EVs<sup>veh-MRC5</sup> and EVs<sup>MMA-MRC5+NAC</sup> groups, $n = 9$ mice for EVs<sup>MMA-MRC5</sup> group, mean ± SEM, two-sided unpaired $t$-test). Data are partially previously represented in Fig. 2g.

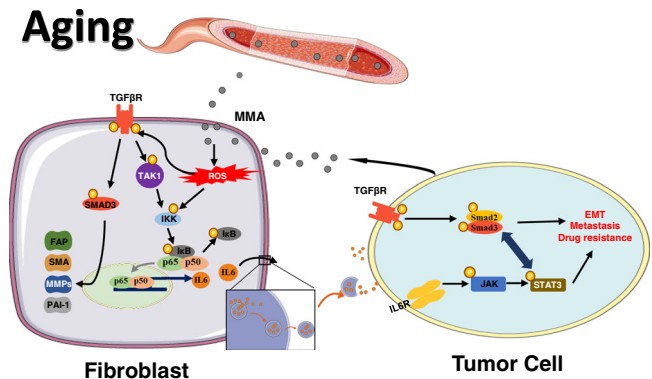

**Fig. 6 | MMA activates fibroblasts and induces their EV-associated IL-6 secretion, which drives metastatic reprogramming in tumor cells.** MMA produced by tumor cells induces generation of ROS in fibroblasts. ROS activates TGFβ signaling, which promotes expression of CAF markers, and NF-κB signaling, which promotes IL-6 loading into and secretion through EVs. EVs loaded with IL-6 activate STAT3 signaling and TGFβ signaling in tumor cells, promoting EMT, drug resistance and metastasis.

## Cell culture treatments

For all time courses signaling experiments, cells were seeded one day before inhibitor and MMA treatments. Inhibitors were added 30 min before any MMA treatments. Cells were treated at the same time, then protein was harvested at different time points. Inhibitors and antioxidants used were as follows: SB431542 (Selleck Chemicals, 5 μM), Cryptotanshinone (Cayman Chemical, 1 μM), IKK16 (Millipore Sigma, 1 μM), Takinib (Cayman Chemical, 10 μM), SkQ1 (Cayman Chemical, 1 μM), NAC (Sigma Aldrich, 2 mM).

For long-term EV treatments of tumor cells (such as for EMT marker measurement and functional assays such as drug resistance, colony formation and in vivo experiments), cells were seeded and then treated with 5 μg/ml of EVs for 3–5 days. For short-term EV treatments of tumor cells (such as for signaling activation measurement), cells were treated with 5 μg/ml of EVs for 3 h.

## Proteomics analysis of extracellular vesicles

EVs were isolated from the conditioned medium of vehicle- or MMA-treated MRC5 fibroblast and total protein amount in EVs was measured as described above. 50 μg of total protein from each sample was TCA precipitated and acetone washed. Pellets were re-suspended in 8 M urea, 50 mM ammonium bicarbonate (AMBIC). Proteins were reduced and alkylated with dithiothreitol and iodoacetamide. Samples were diluted to 2 M urea with 50 mM AMBIC and digested overnight with Lysyl endopeptidase (lysC, Wako Chemicals USA, Inc.), then diluted to 1 M urea and digested with trypsin (Promega V5111) for 6 h. Peptides were desalted on C18 STAGE Tips[38]. Eluted peptides were dried and re-suspended in 5% formic acid.

Mass Spectrometric analysis was performed on a Thermo Orbitrap Fusion mass spectrometer equipped with a FAIMS Pro ion mobility cell and an Easy nLC-1000 UHPLC. Peptides were separated with a gradient of 5–26% ACN in 0.1% FA over 75 min and introduced into the mass spectrometer by electrospray ionization as they eluted off a self-packed 40 cm, 100 μm (ID) column packed with 1.8 μm, 120 Å pore size, C18 resin (Sepax Technologies, Newark, DE). The column was heated to 60 °C. Peptides were detected using a data-dependent method. For each precursor scan in the Orbitrap, we cycled through five FAIMS compensation voltage values (−40, −50, −60, −70, −80). For each, we allowed up to 1 s for selection of the most abundant precursors for HCD fragmentation (35% NCE) and MS/MS analysis in the ion trap. AGC targets of 4e5 and 1e4 were used for MS1 and MS2 scans, respectively. Ions selected for MS2 analysis were excluded from reanalysis for 45 s. Ions with +1 or unassigned charge were also excluded from analysis.

MS/MS spectra were matched to peptide sequences using COMET (version 2019.01 rev. 5)[39] and a composite database containing the 20,415 Uniprot reviewed canonical predicted human protein sequences (http://uniprot.org, downloaded 5/1/2019) and its reversed complement. Search parameters allowed for two missed cleavages, a mass tolerance of 25 ppm, a static modification of 57.02146 Da (carboxyamidomethylation) on cysteine, and a dynamic modification of 15.99491 Da (oxidation) on methionine. Peptide spectral matches were filtered to 1% FDR using the target-decoy strategy[40] and then to 1% protein FDR. Label-free quantification was performed using peptide intensities from the integrated areas under each corresponding extracted-ion-chromatogram peak. Intensities for all peptides mapping to each protein were summed for each sample. Subsequent data processing, including normalization and statistical analysis, was done using Perseus as described[41].

## Colony formation assays

The base layer of agarose gel plates was made by mixing 1.2% agarose gel solution warmed up at 42 °C with 2× medium with 20% FBS in a 1:1 ratio and allowed to set at room temperature. To form the top layer, 0.6% agar and 2× medium with 20% FBS were warmed at 42 °C. Cells were resuspended in this 1:1 mixture (giving 0.3% agar in 1× medium) and allowed to set for 4 h at room temperature. Cells were treated with EVs and media was changed twice a week. After 3-4 weeks, cells were stained using 0.1% crystal violet in 10% ethanol for 10 min, followed by 5× rinses in dH$_2$O. Plates were first scanned and colonies were counted using Image J.

## Drug resistance assays

A549 and A375 tumor cells were treated with EVs for 5 days. Cells were then seeded into 96-well plates in technical triplicates. The next day, the cells were treated with vehicle control (DMSO (0.1%), carboplatin (Cayman Chemical, 0–150 μM), paclitaxel (Cayman Chemical, 0–20 nM), vemurafenib (Selleck Chemicals 0–800 nM), AZD6244 (Selleck Chemicals, 0–200 nM) at various concentrations. Cells were incubated for 3 days and then fixed in 4% paraformaldehyde (Electron Microscopy Sciences) diluted in PBS for 30 min. After the fixative solution was removed, the plates were washed with PBS and stained with 0.1% crystal violet solution for 15 min. The staining solution was removed and the plates were washed five times and allowed to dry at room temperature. Crystal violet staining was eluted using 10% glacial acetic acid and the absorbance at 590 nm was measured using an Envision plate reader (Perkin Elmer).

## Transwell invasion and migration assays

A549 and A375 tumor cells were treated with EVs for 5 days. After trypsinization, cells were counted and resuspended in serum free media supplemented with 250 μg/ml BSA (Sigma-Aldrich) (assay media) at a concentration of 2 × 10$^5$ cells/ml. 650 μl of media with 10% FBS was used as the chemoattractant and added to the bottom chamber of cell culture inserts, and 250 μl of cells in assay media was added to the top chamber of cell culture inserts. For migration assays, Boyden chamber inserts (BD Biosciences, 8 μm pore size) were used, and for invasion assays, BD BioCoat invasion chambers coated with growth factor reduced Matrigel were used. Invasion chambers were prepared according to manufacturer instructions. Cells were allowed to migrate and invade for 24 h, then cells that had migrated to the lower surface of the membrane were fixed in ethanol and stained with 0.1% crystal violet. 10× images of crystal-violet stained cells were captured using a Nikon DS-Fi2 camera, and quantifications were carried out in an automated fashion using Fiji/ImageJ v1.52. In brief, binary images of the area covered by crystal violet-positive cells was generated using thresholding and settings that were appropriate for control samples, and the same settings were used throughout the analysis. The percentage area covered by crystal

violet-positive cells was quantified for each condition, using a minimum of three technical replicates.

## Oxidative stress assays

For short-term ROS quantification, fibroblasts were seeded in 12-well plates. After 12 h, MMA was added at indicated periods of time. Media was removed and cells were washed with PBS. Then, media containing 20 μM of 2,7-dichlorodihydrofluorescein diacetate (DCFH-DA) (Cayman Chemical) was added to the cells and incubated for 45 min. Cells were then washed with PBS, trypsinized and added to an opaque black 96-well plate. Fluorescence signaling was quantified using an Envision plate reader (Perkin Elmer) and normalized to total cell number. For measurement of MDA, cells were prepared and quantified using the Thiobarbituric Acid Reactive Substances Assay Kit (Cayman Chemical) according to manufacturer instructions.

## Metabolite extraction and mass-spec analysis

Tumor cells were seeded in 10 cm dishes and infected with shMUT virus for 3 days. Cells were washed with cold PBS and incubated with 5 ml FBS free DMEM to collect conditioned medium. 48 h later, the conditioned medium was collected and the tumor cells were counted. 4 volumes of 100% methanol were added to 1 volume of CM, then subjected to speed vacuum for about 4 h until the solution was fully evaporated. The pellets were resuspended in 10 μL of 50% MeOH (in H2O). Then, the mixture was spun down, and 10 μL of supernatant was mixed with 50 μL short-chain fatty acids derivatization solution. The resulting mixture was vortexed and incubated at 60 °C for 1 h, then the mixture was centrifuged at $21,000 \times g$ for 20 min, and the supernatant was analyzed using an Agilent 1290 LC system coupled to an Agilent 6530 quadrupole time-of-flight mass spectrometer with a 130 Å, 1.7 μm, 2.1 mm × 100 mm ACQUITY UPLC BEH C18 column (Waters). We used the following solvent system: A: H2O with 0.1% formic acid; B: Methanol with 0.1% formic acid. 20 μL of each sample was injected, and the flow rate was 0.35 mL/min with a column temperature of 40 °C. The gradient for HPLC-MS analysis was: 0–6.0 min, 99.5–70.0% A; 6.0–9.0 min, 70.0–2.0% A; 9.0–9.4 min, 2.0% A; 9.4–9.6 min, 2.0–99.5% A. Peaks were assigned by comparison with authentic standards. Relative metabolite amounts were normalized to cell number.

## shRNA gene silencing

shMUT #1 (TRCN0000049038), shMUT #2 (TRCN0000049040), shIL6 #1 (TRCN0000059203), shIL6#2 (TRCN0000372668), shIL6R #1 (TRCN0000058779), shIL6R #2 (TRCN0000058780), shGFP (TRCN0000072181), shPCCA #1 (TRCN0000078424), shPCCA #2 (TRCN0000078423), shCHUK #1 (TRCN0000194782), shCHUK #2 (TRCN0000199496), shTGFBR1 #1 (TRCN0000194693), shTGFBR1 #2 (TRCN0000196293) (all from Sigma Aldrich) lentiviruses were produced by co-transfection of HEK293T cells with plasmids encoding psPAX2 (Addgene plasmid 12260) and pMD2.G (Addgene plasmid 12259) using X-tremeGene HP (Roche) in accordance with the manufacturer's protocol. Medium was changed 16 h post-transfection and the virus harvested over 72 h, filtered, and used to infect fibroblasts and tumor cells with 8 μg/ml polybrene (Sigma-Aldrich). Selection of resistant colonies was initiated 24 h after selection using 2 μg/ml of puromycin (Sigma-Aldrich).

## qPCR

RNA was extracted from cells using the Ambion PureLink RNA Mini Kit (Life Technologies) according to manufacturer's instructions and treated with DNase I (Amplification grade, Sigma-Aldrich). cDNA was synthesized using the iSCRIPT cDNA synthesis kit (BioRad) and analyzed by qPCR using SYBR green master mix (Life Technologies) on a QuantStudio6 Real-Time PCR system with QuantStudio Real-Time PCR software v1.3 (Life Technologies). Exported data were further

processed in Excel for Office 365. Target gene expression was normalized to expression of TBP and ACTB. Primer sequences are as follows: IL6 forward: ACTCACCTCTTCAGAACGAATTG, IL6 reverse: CCATCTTTGGAAGGTTCAGGTTG. IL6R forward: CCCCTCAGCAATGTTGTTTGT, IL6R reverse: CTCCGGGACTGCTAACTGG.

## Global gene expression analysis (RNA-seq)

RNA was extracted from cells using the Ambion PureLink RNA Mini Kit (Life Technologies) according to manufacturer's instructions and treated with DNase I (Amplification grade, Sigma-Aldrich). Total RNA integrity was checked using a 2100 Bioanalyzer (Agilent Technologies, Santa Clara, CA). RNA concentrations were measured using the Nanodrop system (Thermo Fisher Scientific, Inc., Waltham, MA). Preparation of RNA sample library and RNA-seq were performed by the Genomics Core Laboratory at Weill Cornell Medicine. RNA was prepared using TruSeq Stranded mRNA Sample Library Preparation kit (Illumina, San Diego, CA), according to the manufacturer's instructions. The normalized cDNA libraries were pooled and sequenced on Illumina NovaSeq6000 sequencer with pair-end 50 cycles. The sequencing libraries sequenced with paired-end 50 bps on NovaSeq6000 sequencer. The raw sequencing reads in BCL format were processed through bcl2fastq 2.19 (Illumina) for FASTQ conversion and demultiplexing. After trimming the adaptors with Cutadapt (version1.18)(https://cutadapt.readthedocs.io/en/v1.18/), RNA reads were aligned and mapped to the GRCh38 human reference genome by STAR (Version2.5.2) (https://github.com/alexdobin/STAR)[42], and transcriptome reconstruction was performed by Cufflinks (Version 2.1.1) (http://cole-trapnell-lab.github.io/cufflinks/). The abundance of transcripts was measured with Cufflinks in Fragments Per Kilobase of exon model per Million mapped reads (FPKM)[43,44]. Gene expression profiles were constructed for differential expression, cluster, and principle component analyses with the DESeq2 package (https://bioconductor.org/packages/release/bioc/html/DESeq2.html)[45]. For differential expression analysis, pairwise comparisons between two or more groups using parametric tests where read-counts follow a negative binomial distribution with a gene-specific dispersion parameter. Corrected $p$ values were calculated based on the Benjamini-Hochberg method to adjusted for multiple testing.

## Subcutaneous injections and metastasis formation assay in mice

Tumor cells were infected with a EF1A-Luciferase-p2A-GFP vector prior to EV treatments. Cells were trypsinized and resuspended on ice in a 1:1 of PBS: Matrigel mixture. Female nu/nu athymic mice were anesthetized with isoflurane and injected with 500,000 A549 cells, 10,000 A375 cells or a mixture of 100,000 A549 and 400,000 MRC5 cells, or 1,000,000 A375 cells with different mutation (for IHC) in 100 ul subcutaneously into the left flank. Primary tumor growth was monitored by imaging weekly using the IVIS Spectrum CT Pre-Clinical In Vivo Imaging System (Perkin-Elmer), and luminescence was measured and quantified using Living Image Software v.4.5 (Perkin-Elmer). To visualize metastatic spread, mice were sacrificed at the endpoint of 5 weeks and organs, including livers, lungs, brains, spleens and kidneys, were placed into 12-well plates and imaged using the IVIS Spectrum CT Pre-Clinical In Vivo Imaging System (Perkin-Elmer), and luminescence was measured and quantified using Living Image Software v.4.5 (Perkin-Elmer).

## Statistics and reproducibility

All measurements used for statistical analyses in independent experiments were taken from distinct samples. Data analyses were performed using Microsoft Excel 2013 and GraphPad Prism7. Unless otherwise specified, results are expressed as mean ± SEM. The two-tailed Student's t-test, one-way ANOVA and two-way ANOVA were used to determine significance. For all western blots, experiments were independently repeated n = 3 times and representative images are

shown. Sample sizes for mouse experiments were chosen based on power calculations using expected results. No data were excluded from the analyses. Mice and cells were randomized before allocation into different experimental groups. The investigators were not blinded to allocation during experiments and outcome assessment.

### Reporting summary

Further information on research design is available in the Nature Research Reporting Summary linked to this article.

## Data availability

All data generated or analyzed during this study are included in this published paper (and its supplementary information files) under the Source Data file, or has been previously published as described in the Methods. The RNA-seq data generated in this study have been deposited in the Gene Expression Omnibus (GEO) repository under accession code GSE190929. All raw data files, peak lists, and the sequence database for the proteomics analysis have been deposited in the MassIVE repository (https://massive.ucsd.edu) under ID MSV000090312. All the other data are available within the paper and its Supplementary Information. Source data are provided with this paper.

## Code availability

The quantification of invasion/migration assay images were carried out in an automated way on Fiji/ImageJ v1.52. The scrip was deposited into GitHut (https://doi.org/10.5281/zenodo.7143661).

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

## Acknowledgements

We are grateful to members of the Blenis and Cantley Laboratories for critical input on this project. Elena Piskounova kindly provided luciferase plasmid for xenograft experiments. This research was supported by the following grants: R01GM051405 (J.B.), R01CA46595 (J.B.). C.-J.G. is supported by the AGA Research Foundation, WCM-RAPP Initiative, The. W. M. Keck Foundation, the National Institutes of Health (DP2 HD101401-01). A.L. is funded by R01CA256188-01. V.Luga is funded by Canadian Institutes of Health Research (CIHR) fellowship. The authors acknowledge the assistance of the staff at the Microscopy & Image Analysis Core at WCM. TEM was purchased with funds from an NIH Shared Instrumentation Grant (S10RRO27699).

## Author contributions

Z.L., V.Low, and J.B. conceived and supervised the project. Z.L. and V.Low. performed most of the cell culture, molecular biology experiments, the EMT-related experiments, the invasion and migration experiments, drug resistance and the mouse experiments. V.Low. prepared the RNA for RNA-seq, generated the viral particles for knockdowns, generated the genetically modified cell lines, and performed qPCR analyses. E.E and A.L performed patient single cell data analysis. M.H. performed EVs quantification by Nano Sight. J.S. and V.Luga performed Electron Microscopy analysis of EVs and quantification of particle size. W.J. and C.-J.G. performed mass-spec analysis. N.D. performed proteomic analysis. K.G. assisted in data analysis. B.P., J. E., S.C. and T.S. assisted in some cell culture and molecular experiments. Z.L., V.Low., and J.B. analyzed the data. D.L. and L.C.C. oversaw experiments and provided feedback. The paper was written by Z.L., V.Low., and J.B., and edited by A.L. and V.Luga. All authors discussed the results and approved the paper.

## Competing interests

L.C.C. owns equity in, receives compensation from and serves on the board of directors and scientific advisory board of Agios Pharmaceuticals and Petra Pharma Corporation. The other authors declare no competing interests.

## Additional information

[1]Meyer Cancer Center, Weill Cornell Medicine, New York, NY 10021, USA. [2]Department of Pharmacology, Weill Cornell Medicine, New York, NY 10021, USA. [3]Department of Medicine, Weill Cornell Medicine, New York, NY 10021, USA. [4]Institute for Computational Biomedicine, Weill Cornell Medicine, New York, NY, USA. [5]Department of Biochemistry, Weill Cornell Medicine, New York, NY 10021, USA. [6]Departments of Pediatrics, and Cell and Developmental Biology, Weill Cornell Medicine, New York, NY 10021, USA. [7]Jill Roberts Institute for Research in Inflammatory Bowel Disease, Department of Medicine, Weill Cornell Medicine, New York, NY 10021, USA. [8]Department of Physiology and Biophysics, Weill Cornell Medicine, New York, NY 10021, USA. [9]Present address: Department of Radiology, Memorial Sloan Kettering Cancer Center, New York, NY 10065, USA. [10]Present address: Dana Farber Cancer Institute, Boston, MA 02215, USA. [11]These authors contributed equally: Zhongchi Li, Vivien Low. ✉e-mail: job2064@med.cornell.edu

