## [Peer Review File · Nature Communications]

This manuscript has been previously reviewed at another journal that is not operating a transparent peer review scheme. This document only contains reviewer comments and rebuttal letters for versions considered at *Nature Communications*.

Reviewer Response

Reviewers' expertise:

Reviewer #1: Lung cancer and metabolism

Reviewer #2: Extracellular vesicles in cancer and metastasis

Reviewer #3: Cancer-associated fibroblast biology

Reviewer #1 (Remarks to the Author):

In their manuscript, Li et al describe the impact of MMA treatment on fibroblasts and the subsequent ability of vesicles produced by these fibroblasts to impact cancer cell migration and metastatic potential. The current study builds on the authors' prior work establishing MMA as a byproduct of propionate metabolism whose presence in the plasma is a function of vitamin B12 deficiency which increases as patients age. In that study, authors provided evidence that MMA induces EMT and increases metastatic potential in cancer cell models via activation of TGF- β and SOX4.

In the present study, authors provide evidence that tumor expression of propionate pathway enzymes correlates with EMT marker expression. Induction of MMA production in cancer cells in vitro results in the secretion of factors which induce CAF phenotypes in co-cultured fibroblasts, similar to direct MMA treatment. Isolated vesicles produced by these CAFs can then stimulate cancer cells to adopt phenotypes associated with EMT: migratory nature, increased metastatic potential, and chemotherapy resistance. Proteomic analysis of CAF secreted vesicles from MMA treated fibroblasts reveals increased production of IL-6. Cancer cells receiving these CAF secreted vesicles induce STAT3 and SMAD signaling, and suppression of these pathways reverses the effects of CAF vesicle treatment. CAFs treated with MMA exhibit increased TGF- β and NF- κ B signaling. Finally, the authors then show that MMA treatment stimulates ROS production in fibroblasts and anti-oxidant treatment reverses CAF formation, IL-6 secretion, and EMT markers in vesicle treated cancer cells. The authors conclude that elevated MMA levels, present in aged individuals, can alter tumor stroma in a way that promotes cancer cells EMT and metastasis via an IL-6-mediated mechanism.

The data presented is technically well done. Overall, the interest primarily hinges on believing that the induction of ROS by MMA treatment in fibroblasts in culture has physiological relevance. The remaining signaling is largely supported by existing literature, although some control experiments to shore up these findings in this system would also be useful. Specifically:

1) The authors rely heavily on exogenous MMA added to fibroblasts to induce phenotypes in these cells. Given that ROS and downstream stress responses can be triggered somewhat non-specifically, it would be helpful if at least some of the phenotypes in CAFs (ROS induction and TGF- β /NF- κ B signaling) were reproduced with "physiological" MMA, such as from sera of aged donors, as in their prior work.

We thank the reviewer for this comment. At the time of initial submission, we had a manuscript under review (now published at *Nature Metabolism*) that provided much of the physiological premise for physiologically high concentrations of MMA in the tumor microenvironment, largely produced by tumors (Gomes et al., 2022). This study

demonstrated that aggressive tumor cells dysregulate propionate metabolism to accumulate high concentrations of MMA, driving cancer progression, leading to the question addressed in our current manuscript- how do these high levels of MMA in the tumor microenvironment affect the stroma?

We agree with reviewers that reproducing experiments with physiological MMA would strengthen our study. However, we have elected not to use aged serum as physiological MMA, as there are many factors in aged serum, both metabolic and non-metabolic, that can affect a panoply of different functions in CAFs. In our *Nature* paper, we demonstrated that serum MMA is largely encapsulated in extracellular vesicles, and in our recently published *Nature Metabolism* study, we demonstrate that tumor cells with altered propionate metabolism, such as *MUT* knockdown, secrete higher levels of MMA (Gomes et al., 2022; Gomes et al., 2020). Therefore, to reproduce the results observed with exogenous MMA by “physiological MMA”, we utilized extracellular vesicles secreted by tumor cells with knockdown of *MUT*.

We confirmed that EVs from A549 lung cancer cells with shMUT (shMUT-A549-EVs) contain higher levels of MMA than control EVs from A549 cells with shGFP (shGFP-A549-EVs) (Figure s1e). When we treated MRC5 fibroblasts with these EVs, fibroblasts that had been treated with shMUT-A549-EVs displayed increased expression of CAF markers compared to fibroblasts treated with shGFP-A549-EVs (Figure s1f). Additionally, shMUT-A549-EVs were able to activate TGFbeta/NF-kB signaling and ROS induction, reproducing the effects of treatment by exogenous MMA (Figure s6a, s7a).

We hope that in addition to our recent publication in *Nature Metabolism*, these additional experiments, which demonstrate that tumor-produced MMA is encapsulated in vesicles and can lead to ROS generation, activate the TGF-beta and NF-kB signaling pathway and activate CAF markers, in addition to the recent publication of our *Nature Metabolism* paper, will provide sufficient evidence that physiological MMA can indeed activate CAFs to promote tumor progression.

2) The connection between MMA and ROS is not particularly well explored. Do exogenous factors that induce ROS stimulate a CAF-like state or is there something specific about MMA?

We have looked at the ability of other exogenous factors that induce ROS to promote CAF activation, and found that even though they can induce a similar ROS level, they cannot fully reproduce the effect of MMA on fibroblasts (Figure s7d-e). This suggests

MMA may increase ROS through a specific mechanism and/or that MMA activates other processes that work with ROS to induce activation of fibroblasts. We have begun to follow up on several leads on how MMA generates ROS in CAFs, although we consider this an extensive amount of work that goes beyond the scope of this report.

3) Is IL-6 addition itself sufficient to induce a response in cancer cell models that is similar to CAF EVs?

This question is an important one that we had considered and addressed, but omitted from the original submission due to scope and space. When we added IL-6 alone into the medium, we found that it could not produce the effect of MMA-EVs (EVs isolated from fibroblasts treated with MMA) at any concentration; however, when we added IL-6 directly in combination with veh-EVs (EVs isolated from fibroblasts treated with vehicle), we saw activation of JAK-STAT signaling and induction of EMT markers (Figures shown below). This indicates that there may be a synergistic effect between IL6 and other components of the EVs. There are reports of similar effects with other factors, for example, EV-associated TGFβ1 results in more prolonged cellular signaling compared to free TGFβ1 (Shelke et al., 2019). We are happy to include these data in the revised manuscript, if requested.

(Left): Immunoblots measuring signaling activation of A549 cells treated with varying concentrations of IL-6, alone or in combination with veh-EVs or MMA-EVs from MRC-5 fibroblasts for 3 hours. (Right): Immunoblots measuring EMT marker expression of A549 cells treated with varying concentrations of IL-6, alone or in combination with veh-EVs or MMA-EVs from MRC-5 fibroblasts for 5 days.

4) In Figure 3, the authors argue that SB, Cry, or IL6R suppression blocks MMA-EV stimulated changes in migration. However, they do not assess the effects of these treatments in control cells. Are the effects of SB, Cry, or IL6R suppression specific to MMA-EV or do they generally inhibit migration/invasion?

We had originally omitted this data in the original submission in the interest of space, and include the data below. SB and Cry treatment, as well as IL6R suppression, do not

decrease migration or invasion significantly in control cells (Figures shown below). We are happy to include these conditions in the figures if requested by reviewers.

(Left): IL6R or GFP was knocked down in A549 tumor cells. Invasion and migration transwell assays were performed 4 days after the infection. (Right): A549 tumor cells were treated with TGFβR inhibitor SB431542 or the stat3 inhibitor cryptotanshinone for 4 days, and then the invasion and migration transwell assays were performed.

The authors might also consider:

At times it is quite confusing from looking at the figures to discern whether a treatment is being applied to a cancer cell or fibroblast, and whether a protein lysate is being taken from a cancer cell or fibroblasts. It would be helpful if the authors had some scheme to more clearly indicate this on the figures themselves.

We thank the reviewer for pointing this out. We have adopted a better labeling system following the format “*compartment*^{manipulation-cell line} for example, “CM^{shMUT-A549}” and “EVs^{MMA-MRC5}”, and updated the names in the manuscript and figures. Additionally, we have added in a schematic outlining the general experimental structure for reference (Figure 2c).

Reviewer #2 (Remarks to the Author):

In this work, Blenis and collaborators are expanding on their paradigm-shifting discovery of methylmalonic acid (MMA) as an oncometabolite that is increased in aging and acts to upregulate the mesenchymal markers/characteristics in cancer cells and promote the epithelial-to-mesenchymal transition (EMT) by inducing TGFβ and upregulating Sox4 transcription factor (Gomes A. et al, 2020, Nature). Here, authors demonstrate that MMA can be also produced by malignant cells and can act upon the tumor microenvironment (TME). They also report that:

- MMA-exposed fibroblasts become activated into cancer-associated fibroblasts (CAFs)
- These MMA-treated CAFs produce the extracellular vesicles (EVs), which carry IL-6 and are involved in the stimulation of the TGF β pathway
- TGF β and IL-6, in turn, drive the EMT in malignant cells thereby increasing their metastatic potential and resistance to anti-cancer drugs.

Whereas some of presented data are interesting, this work falls short in a number of aspects including novelty and the rigor of proof.

First, giving known ability of MMA to stimulate the TGF β pathway and the importance of this pathway as well as IL-6 in the generation and pro-tumorigenic activities of CAFs (, presented here finding are not surprising and well supported by existing literature. The role of CAF-produced EVs in EMT and in increasing the aggressive growth and metastasis of cancer cells has also been very well documented (for example, see review from Conigliaro & Cicchini, 2020, J Clin Med). This substantial prior body of published work seriously reduces the novelty of presented here manuscript, which is perhaps better suited to a more specialized journal.

We thank the reviewer for these comments, and apologize that the manuscript writing may have been confusing or unclear on certain points. While we have previously shown that MMA can stimulate the TGFbeta pathway in tumor cells as the reviewer points out, here we show for the first time that MMA-induced stimulation of the TGFbeta pathway also occurs in fibroblasts and leads to their activation, which results in a feed-forward mechanism to further promote carcinogenesis. Additionally, we do not discuss the role of IL-6 in the generation of CAFs but the role of CAF-generated IL-6 to promote tumor progression.

It is true that the effects of CAF-produced EVs is well-studied. However, due to the many different kinds of CAFs, with different characteristics induced by different factors and under different contexts, we believe that our demonstration of a new aging- and tumor-regulated metabolite as an inducer of CAFs is a novel finding. Additionally, although IL-6 has been shown to exist in EVs, we provide the first demonstration that EV-associated IL-6 is functional in tumor progression. Finally, this is also the first demonstration that MMA can act on non-tumor cells in cancer.

With the additional experiments provided during revision, our novel findings include:

1. MMA, produced in tumor cells with altered propionate metabolism, is secreted in EVs.
2. The MMA-containing EVs from tumor cells or MMA added directly to fibroblasts, promotes their activation to CAFs.
3. MMA-dependent activation of CAFs depends on stimulation of canonical and non-canonical TGF-beta signaling and requires MMA-dependent increase in ROS plus yet to be uncovered processes, whereas interestingly, ROS induction by other agents does not.
4. Once activated, CAFs secrete EVs that then feedback on the MMA-secreting tumor cells to further promote cancer cell progression and chemoresistance.
5. IL6, only when associated with CAF produced EVs but not when added directly, is required for the tumor progression (Figures shown below).

(Left): Immunoblots measuring signaling activation of A549 cells treated with varying concentrations of IL-6, alone or in combination with veh-EVs or MMA-EVs from MRC-5 fibroblasts for 3 hours. (Right): Immunoblots measuring EMT marker expression of A549 cells treated with varying concentrations of IL-6, alone or in combination with veh-EVs or MMA-EVs from MRC-5 fibroblasts for 5 days.

Second, although largely compelling, presented data tell us what CAN happen in the TME but not what really DOES happen. I am convinced that MMA treatment *in vitro* can induce activation of CAFs and subsequent EMT. However, I am not convinced that MMA indeed is responsible for activation of CAFs and subsequent EMT in the TME of growing tumors. Furthermore, the importance of events elicited specifically by MMA (versus many others well documented stimuli) remains unclear.

We understand the reviewer's concerns, considering that at the time of initial submission, we had not disclosed that many of the reviewer's concerns regarding the physiological relevance of what happens in the TME were already addressed in a separate manuscript that was at the time under review, but is now published in *Nature Metabolism* (Gomes et al., 2022). That study provided the physiological premise for high physiological concentrations of MMA in the tumor microenvironment, largely produced by tumors. In the paper, we demonstrated that ERK2 or TGF β induces alterations in propionate metabolism, resulting in increased production of MMA and EMT, and that cancer cells isolated from aggressive tumors also display dysregulated propionate metabolism, resulting in the accumulation of high concentrations of MMA, driving cancer progression. We hope that, now that our previous manuscript is published, the reviewer will find that many of their concerns have been addressed.

That said, to strengthen the evidence that tumor-produced MMA contributes to these cancer-promoting events *in vivo*, we have performed additional experiments. We knocked down GFP or PCCA in A375 cells, which have high endogenous production of MMA, and implanted these cells into mice. Using immunohistochemistry, we found that there were indeed infiltration and activation of activated fibroblasts into tumors, and that this was significantly impaired by disruption of MMA production through PCCA knockdown (s1a-c). We hope the reviewer will agree that this demonstrates that MMA produced by tumors does indeed contribute to activation of fibroblasts in the tumor microenvironment.

Specific points

1. The Discussion section is not balanced. Previous studies have shown that inflammation and MMA are increased in aged humans – do aged humans harbor activated CAFs in their tissues? While malignancies are indeed prevalent in aged populations, some cancers actually have better prognosis with age. How is that explained from the MMA-centric point of view? It would be great to discuss these and other issues similarly standing out when hypothesizing about an overriding impact of MMA.

We thank the reviewer for this comment. Indeed, it has been shown that older people have increased tissue fibrosis (Murtha et al., 2019). While we do not argue that this increased tissue fibrosis is due to MMA, the possibility is certainly intriguing. However, while we have previously shown that MMA is an important age-induced factor contributing to poorer prognoses of cancers, the current manuscript is based on the premise that high levels of MMA produced by tumor cells (which can be exacerbated by high levels of MMA in aged serum) drives CAF generation in the tumor microenvironment. We apologize as this point was likely not clear because this manuscript builds upon another manuscript that was under review at the time of initial submission, but is now published with *Nature Metabolism* (Gomes et al., 2022). The previous paper shows that aggressive tumors dysregulate propionate metabolism to produce high levels of MMA (some excerpts shown below), which we now show with additional experiments is secreted in the tumor microenvironment in the form of EVs (Figure s1e-f, s6a, s7a).

Figures from (Gomes et al., 2022)

The reviewer is correct that some cancers actually have better prognosis with age, although these are the exception rather than the rule (de Magalhaes, 2013). Our current manuscript focuses on lung cancer and melanoma, wherein there is strong evidence that aging is closely related to tumor progression. While we do not focus on the role of aging in this paper, but rather MMA as a tumor produced messenger activating fibroblasts in the tumor environment, we do agree that these are interesting questions, and based on our previously published study regarding aging-induced MMA, we have added in a brief discussion of the link between age and tissue fibrosis. Additionally, in no way are we claiming that MMA will have the same effect on all tumor cells, and cannot make that claim until we study more cancers.

2. The importance of presented here MMA-driven mechanism versus other mechanisms underlying the activation of CAFs is not clear. Authors correlated mRNA levels of genes that act to prevent MMA accumulation with those of mesenchymal signature, but this is just a correlation. Does a cohort of specific human cancers present a greater level of MMA compared to normal tissues? Do those with greater levels of MMA indeed harbor a greater proportion of inflammatory CAFs?

We thank the reviewer for this comment, and apologize once again for omitting our other, now published study in our initial submission, which addresses many of these points. In the *Nature Metabolism* study, we demonstrate that more aggressive cancers have higher levels of MMA, and this is due to dysregulation of propionate metabolism (Gomes et al., 2022).

Figures from (Gomes et al., 2022)

Knockdown of genes downstream of MMA in the propionate metabolism pathway, and overexpression of genes upstream, effectively change MMA levels and influence tumor aggressiveness. Of note, knockdown of the genes *MUT*, *MCEE*, *MMAB*, which we show in the current manuscript to be inversely correlated with a mesenchymal signature, were all already shown to increase MMA levels in the now published study (Gomes et al., 2022).

Additionally, we have unpublished data, which will be used for a future, separately manuscript, showing that lung tumor tissues of patients do indeed have higher levels of MMA compared to matched normal lung tissues in patients (Figure shown below, right). Finally, we looked at the TCGA dataset of 501 lung squamous carcinoma tumors, and found a correlation between low *MUT*, *MCEE*, *MMAB* and *MMAB* levels (indicating high MMA) and high expression of cancer-associated fibroblast markers *ACTA1* (encoding for α -SMA) and *FAP* (Figure s1d), suggesting that tumors with greater levels of MMA do, indeed, harbor a great proportion of inflammatory CAFs.

MMA levels in tumor compared to matched non-malignant tissue, from 20 resected lung adenocarcinoma and squamous cell carcinoma patients.

s1d

TCGA Lung Squamous Carcinoma

3. Furthermore, are there any animal models recapitulating these relationships without using cell lines artificially engineered to produce more MMA? Is there evidence that unequivocally supports the role of malignant cell-derived MMA in developing aggressive TME in the models where fibroblasts become CAFs within the TME and are not treated with MMA ex vivo? Answering these questions will help to exclude potential artifacts associated with MMA treatment of cultured cells in vitro. Experiments that somehow inactivate MMA production by malignant cells might be required to address these important questions.

We thank the reviewer for these suggestions. In order to address these questions, we knocked down *GFP* or *PCCA* in A375 cells, which have high endogenous production of MMA, and implanted these cells into mice. Using immunohistochemistry, we found that there was indeed infiltration and activation of activated fibroblasts into tumors, and that this was significantly impaired by disruption of MMA production through *PCCA* knockdown (Figure s1a-c).

s1a

s1b

s1c

Additionally, we have performed experiments where we show that the malignant-cell derived MMA is largely encapsulated in lipid vesicles, and that these lipid vesicles can reproduce the phenotypes observed in CAFs by MMA (Figure s1e-f, s6a, s7a).

We hope that in addition to the now-published manuscript, the reviewer will find that these additional experiments provide strong evidence for the role of MMA produced by malignant cells in developing an aggressive TME.

4. Authors conclude that MMA can shape the TME. However, all tumor growth experiments are carried out in the immune-compromised hosts. This design prevents an important consideration towards the immune compartment of the TME. For example, immune cells infiltrating the tumors might be responsible for producing a notably proportion of the intratumoral IL-6. Furthermore, CAFs-generated EVs are known to dramatically suppress the anti-cancer immunity. The importance of MMA-elicited events in the immune-competent models must be experimentally addressed.

We thank the reviewer for these comments, and we agree that the immune compartment of the TME is an interesting and important aspect of tumor progression. However, we believe this is outside the scope of our paper. Furthermore, the effect of MMA on immune cells is currently being studied by the Gomes lab at Moffit Cancer Center, with our blessing. The use of immune-compromised hosts is common practice when doing human xenografts to prevent rejection of these cells.

5. The Abstract contains grossly misleading statements such as “methylmalonic acid (MMA), an aging-increased oncometabolite that is also produced by aggressive cancer cells, activates fibroblasts in the TME and stimulates the release of extracellular vesicles (EVs)”. First, authors do not provide evidence that more aggressive cancers produce more MMA compared to less aggressive. Second, Figure S2a supports the description in the text (“We did not observe a significant difference in the number or size of extracellular vesicles secreted by MMA-treated fibroblasts (MMA-EVs) compared to those secreted by vehicle-treated fibroblasts (veh-EVs).”) – but not the statement in the Abstract that MM stimulates the release of EVs.

We thank the reviewer for pointing out our oversight, and apologize once again for omitting our other, then under-revision and now published manuscript, in which we demonstrate that aggressive cancers produce more MMA compared to less aggressive cancers, in the original submission for reviewers. This current manuscript was written under the assumption that the other manuscript would be published first, but we did not consider how reviewers may miss much of the foundation of this paper without it. Some figures from that paper demonstrating that aggressive cancers have higher levels of MMA is shown below. Additionally, we showed in that paper that EMT inducers, such as TGF β and TNF α , can also stimulate MMA production by altering propionate metabolism (Gomes et al., 2022).

Figures from (Gomes et al., 2022)

As for the second statement regarding the release of EVs, we apologize for that oversight and have corrected our wording in the abstract.

6. Figure 1. While MMA indeed stimulates the expression of CAF markers in MRC-5 and BJ fibroblasts in vitro and their ability to support tumor growth vivo, this does not mean that formation of CAFs in the growing tumor depends on MMA or even that MMA contributes to this process.

We thank the reviewer for this comment and want to emphasize that we do not and cannot claim that MMA is essential for CAF formation. We only claim that among the many inducers of CAFs in the TME, MMA is one of them, and important in certain types of cancer. However, to strengthen this evidence, we performed additional experiments where we knocked down *GFP* or *PCCA* in A375 cells, which have high endogenous production of MMA, and implanted these cells into mice. Using immunohistochemistry, we found that there was indeed infiltration of activated fibroblasts into tumors, and that this was significantly impaired by disruption of MMA production through *PCCA* knockdown (Figure s1a-c).

7. Figure 2. It is not clear why authors right away focused on EVs versus soluble factors that could be upregulated by MMA treatment. Was it because of previously published reports linking CAF-EVs to EMT, drug resistance and metastasis? Why were these reports not cited and properly discussed in the context of present findings?

We apologize to the reviewer that we did not make our reasons for focusing on EVs versus soluble factors more clear, as we had in fact already addressed these reasons in the Figure 2a, 2c and s3a, which we have included again below. While the conditioned media of MMA-treated fibroblasts could induce EMT in the cancer cells, as well as the isolated EVs extracted from this conditioned media, the supernatant of this conditioned media after the EVs were removed could no longer induce EMT in these cells (Figure 2a, 2d, s3a). This clearly demonstrated to us that the active factor inducing EMT was contained in the EVs, and not among the soluble factors in the conditioned media. As the reviewer suggested, we looked at EVs versus soluble factors in the first place because of previously published reports regarding the importance of EV as signaling mediators, and had cited these reports in the manuscript: "EVs are loaded with signaling molecules and genetic material, and function as essential signaling mediators in the tumor microenvironment (Luga et al., 2012; Wendler et al., 2017)." We hope that in highlighting this text and the experiments, the reviewer will be satisfied as to why we decided to focus on EVs.

8. Figure 3 – results are overinterpreted - these data do not exclude alternative interpretation. While it is possible that STAT3 activation in response to MMA-CAF-EVs is indeed due to IL-6 transferred by these vesicles, it also could be because of induction of malignant cell-derived IL-6 (or, for that matter, of many other possible STAT3-activating cytokines) in response to these EVs. If authors are indeed set on proving their hypothesis, they should be thinking of modulating IL-6 levels in MMA-CAFs, so they load less of this cytokine on EVs.

We apologize that we did not make our supplemental Figure 5, which already addresses these questions, more obvious for the readers of our manuscript. We did in fact modulate IL-6 levels in MMA-treated CAFs, as shown in all of Figure S5. We confirmed that this resulted in reduced loading of IL6 into EVs, and measured the ability of MMA-EVs with IL6 knocked down to activate signaling, induce EMT, modulate drug resistance, invasion and migration. We have included the figure again below for the reviewers' convenience.

S Figure5

9. Likewise, while the role of TGFβ and STAT3 in CAFs activation is pretty clear, data that rely on the use of pharmacologic agents of dubious specificity (Figure 4-6) and link MMA-induced ROS with activation of TGFβ-driven stimulation of the TAK1-IKK-NF-κB pathway should be rigorously supported by clean and unequivocal genetic studies.

We thank the reviewer for these questions, and have performed additional genetic studies with knockdowns of TGFBR1 and IKK1, looking at their effects on CAF markers. These studies support our findings from using pharmacological agents (Figure s6b-e).

Reviewer #3 (Remarks to the Author):

The authors argue that methylmalonic acid MMA is an aging-increased oncometabolite produced by aggressive cancer cells. The authors argue that methylmalonic acid secreted by melanoma cells activates fibroblasts in the tumor microenvironment and stimulates the release of specific proteins in extracellular vesicles. These extracellular vesicles in turn provide IL-6 to the tumor cells, activating the JAK/STAT3 and TGF- β signaling pathways in the tumor cells and inducing EMT, drug resistance and metastasis.

Major comments

The finding that a metabolite, MMA, can induce fibroblasts to express CAF markers, to secrete specific cytokines in EVs, and to affect important properties of tumor cells is an interesting finding. A role for this metabolite in cancer-CAF interaction hasn't been previously reported. There are parts of the pathway that the authors propose that they have demonstrated relatively convincingly. The effects of MMA on fibroblasts are laid out clearly and the reversal with the inhibitors is also clear. Some aspects of the model are less convincing at this point and additional controls are needed to determine whether these effects are mediated through a different CAF state, a different EV state, or making a higher fraction of fibroblasts CAF like.

We thank reviewer #3 for their positive review of our manuscript.

Comments

1. Previous studies have shown that CAFs secrete IL-6 in EVs and TGF β in EVs to affect cancer cells and these should be cited. (e.g., Goulet et al BMC Cancer 2019, Liu et al, Exp Cell Res 2020).

We thank the reviewer for these suggestions. The first paper by Goulet et al does report that CAFs can secrete IL6 to regulate tumor EMT, but does not show that IL6 is EV-associated. Importantly, we have data demonstrating the effects of CAF-produced IL6 on tumor cell progression requires its presentation with EVs as IL6 alone, at the concentrations used, does not lead to signaling activation or EMT (See below). This data is not currently included in the manuscript, but we can include it if the reviewer finds it appropriate. The second paper, by Liu et al, reported that CAFs can secrete TGF β 1 through EVs and in our data, we did not see a significant difference in the levels of TGF β 1 between the EVs isolated from MMA- and vehicle- treated MRC5 fibroblasts.

(left) Immunoblots measuring signaling activation of A549 cells treated with varying concentrations of IL-6, alone or in combination with veh-EVs or MMA-EVs from MRC-5 fibroblasts for 3 hours. (right) Immunoblots measuring EMT marker expression of A549 cells treated with varying concentrations of IL-6, alone or in combination with veh-EVs or MMA-EVs from MRC-5 fibroblasts for 5 days.

2. In Fig 1d. What is the absolute MMA concentration in the conditioned medium in 1d and how does it compare to the levels in Fig 1g? What is the range of MMA concentration level expected from the change in MUT and other enzymes actually observed in the different cells in Fig 1a?

The concentration of MMA in the conditioned medium ranges from 0.5-5uM, which is much lower than what is used in 1g. This is because MMA in the conditioned media, as well as human serum, is largely encapsulated in lipid particles which facilitate cell entry, allowing much lower concentrations of encapsulated MMA to have the same effect as higher concentrations of exogenously added MMA. In our previous Nature paper, we found that serum MMA is largely packaged in lipid structures, and that using artificial lipid vesicles loaded with MMA allowed us to use much lower concentrations of MMA (1uM) compared to adding MMA directly (1mM) (Gomes et al., 2020). In the current manuscript, we found that MMA was present in the extracellular vesicles of tumor cells, and that these vesicles can activate CAF marker expression. Additionally, the EV-encapsulated MMA makes up most of the effective MMA in the conditioned media in the serum, the supernatant after EVs are removed can no longer activate CAF markers (Figure s1e-g).

While we are unable to measure the changes in MMA levels in the patient data from Figure 1a, we recently published a manuscript in *Nature Metabolism* (under revision at the time of initial submission) which addresses this question. In that study, we demonstrated that aggressive tumor cells dysregulate propionate metabolism to accumulate high concentrations of MMA, driving cancer progression. When we modified various enzymes of the propionate metabolism pathway, including those in Figure 1a,

and measured relative changes in MMA levels, they changed as we expected (see below) (Gomes et al., 2022).

(Gomes et al., 2022).

3. Fig 1 c-g. If the authors deplete MMA from tumor cell conditioned medium, does tumor cell conditioned medium still form CAFs?

We thank the reviewer for this question. Because it is technically challenging to specifically deplete MMA from tumor cell conditioned medium, we inhibited MMA production by tumor cells through knockdown of *PCCA*. When these cells were injected into mice, we saw lower activation and infiltration of fibroblasts into the tumor (Figure s1a-c).

Additionally, we performed experiments demonstrating that MMA secreted by *MUT*-knocked down tumor cells is largely packaged in EVs from tumor cells (Figure s1). These MMA-containing EVs were sufficient to induce CAF formation, while depletion of these vesicles from the conditioned media of *MUT*-knocked down cells abolished its ability to induce CAF markers in fibroblasts (Figure s1-6).

4. Fig 1b is from primary lung cancer showing a change with the transition from epithelial to mesenchymal. Figs 1c to 1g include lung and melanoma cell lines. Melanoma is a surprising choice for the cell line since it is not epithelial in origin and the argument seems to involve epithelial to mesenchymal transition specifically.

While melanoma is not necessarily epithelial in origin, there are many reports that show that melanomas can exhibit epithelial differentiation. The A375 melanoma cancer cell line, for example, is described as having an epithelial morphology on the ATCC website, and can undergo EMT-like changes to become more aggressive (Bao et al., 2020; Cordaro et al., 2017; Laurenzana et al., 2015).

5. The RNA seq suggests that there will be more MMA secreted by more mesenchymal lung cancer cells. Is this true? If so, is it mediated through MMA? There are no follow-up studies about this point.

We apologize for failing to mention that at the time of initial submission, we had a separate manuscript under revision, now published in *Nature Metabolism*, that addresses these questions. In this paper, we showed that mesenchymal tumor cells induced by TGF β /TNF α produce more MMA through dysregulation of propionate metabolism (Gomes et al., 2022) (Figures shown below). This paper highlights a positive feedback loop of tumor progression, wherein more mesenchymal/aggressive tumor cells drive MMA production, which can then further drive their aggressiveness.

Figures from (Gomes et al., 2022)

6. What do the MMA-induced CAFs look like? An image would be helpful. What fraction of fibroblasts have taken a CAF phenotype? Does the MMA increase the fraction of cells that express α-SMA, or do the cells that express α-SMA express it more strongly?

We have the images of these fibroblasts (as shown in the figure below, MRC5 and BJ fibroblasts were treated with MMA for 5 days). Normally, the mature fibroblasts exhibit a thin, wavy, and small spindle morphology, CAFs are often described as immature fibroblasts and appear as large, plump spindle-shaped cells with prominent nucleoli (Gascard and Tlsty, 2016). Vehicle-treated fibroblasts are more interactive and organized. MMA-treated fibroblasts grow in a less cohesive manner and more disorganized morphology.

We have also measured the αSMA level and distribution using immunofluorescent staining (as shown in the figure below). Our data shows that MRC5 fibroblasts express more αSMA under MMA treatment.

7. If the authors would like to rule out pH as a mechanism for MMA's functions, then it seems important to measure pH levels with and without MMA and the other metabolites tested.

We agree that it is important to measure pH levels, and we did in fact do this before starting our experiments. EMEM with 10% FBS, 1% PenStrep, and 25mM HEPES with various acidic metabolites were kept in the incubator for 10 minutes after addition of metabolites, then pH was measured using an electronic pH meter (VWR Symphony B10P Benchtop pH Meter). The results are shown below. 25mM HEPES prevented large changes in pH, and as shown in Figure s1j, other acids that affected pH similarly to MMA were unable to activate CAF markers.

Medium	pH
EMEM	7.28
EMEM+1mM MMA	7.24
EMEM+1mM Succinic acid	7.23
EMEM+1mM propionic acid	7.27
EMEM+1mM malonic acid	7.2

8. How does the effect of MMA on inducing a CAF-like phenotype compare to the effect of cancer cell conditioned medium? How does it compare to other known CAF inducers? Some additional controls would be helpful.

We thank the reviewer for this suggestion, and have performed the experiment requested. The effect of MMA on inducing a CAF-like phenotype is similar to that by cancer cell conditioned medium, as well as TGF β . The effect of other CAF inducers, EGF, TNF α and chemotherapeutic drugs carboplatin and paclitaxel, was less pronounced than with MMA (Figure s1h).

MRC5 fibroblasts were cultured with 1mM MMA, tumor cell conditioned medium, 5ng/ml TGFβ1, 5ng/ml EGF, 5ng/ml TNFα, 20μM carboplatin, 5nM paclitaxel for 5 days.

9. Images in 2e are not visible

We thank the reviewers for pointing this out. We have simply removed the images, as the quantified data is sufficient and the colonies are too small to be seen clearly.

10. Fig 3a: If this experiment were repeated, would the same proteins be identified again? If a similar plot was made for MMA vs MMA EVs or CTRL vs CTRL Evs based on different biological replicates, would there be fewer differences? Would there be any?

We apologize that this figure may have been confusing to understand. In this figure, we performed proteomics on extracellular vesicles derived from MMA-treated CAFs or vehicle-treated CAFs.

11. Fig 3a: are these studies with EV treatment normalized for the amount of cells or the amount of Evs?

We apologize that this figure may have been confusing to understand. This figure does not show EV treatment- it is proteomics performed on the EVs themselves. It is normalized to the amount of total protein in the EVs. There was no significant difference in the protein amount of EVs isolated under the conditions described.

12. Fig 3b: Is it possible some MMA from the culture was acting on the tumor cells and the effect reflects the MMA rather than the EVs? Same with other figures, including the mouse models. Could the MMA itself be having a direct effect separate from the fibroblasts and EVs?

We have measured MMA levels in the EVs isolated from the conditioned media of fibroblasts, and found no difference between the vehicle-treated and the MMA-treated groups, suggesting that MMA is not repackaged into extracellular vesicles produced by fibroblasts under our conditions and acting upon tumor cells (see below).

13. The in vivo mouse studies of metastasis are the most important endpoint because they are in vivo. At present, there are metastasis studies that test the role of ROS by adding NAC. However, the authors don't test the effects of IL-6, or TGF- β , NF- κ B or TAK inhibitors on the development of primary tumors or metastases.

We thank the reviewer for these suggestions. We feel that this complete series of mouse experiments, in addition to what we have provided and will include as described above, would be excessive. We hope that the reviewer will be satisfied with the scope of additional experiments we have added.

14. The authors have few references for their statements about the role of CAFs in the introduction. More references to the papers that established these functions

We thank the reviewer for the suggestion and have included references for our statements on the role of CAFs in the introduction.

References:

- Bao, Y., Ding, Z., Zhao, P., Li, J., Chen, P., Zheng, J., and Qian, Z. (2020). Autophagy inhibition potentiates the anti-EMT effects of alteronol through TGF- β /Smad3 signaling in melanoma cells. *Cell Death Dis* **11**, 223.
- Cordaro, F.G., De Presbiteris, A.L., Camerlingo, R., Mozzillo, N., Pirozzi, G., Cavalcanti, E., Manca, A., Palmieri, G., Cossu, A., Ciliberto, G., *et al.* (2017). Phenotype characterization of human melanoma cells resistant to dabrafenib. *Oncol Rep* **38**, 2741-2751.
- de Magalhaes, J.P. (2013). How ageing processes influence cancer. *Nat Rev Cancer* **13**, 357-365.
- Gascard, P., and Tlsty, T.D. (2016). Carcinoma-associated fibroblasts: orchestrating the composition of malignancy. *Genes Dev* **30**, 1002-1019.
- Gomes, A.P., Ilter, D., Low, V., Drapela, S., Schild, T., Mullarky, E., Han, J., Elia, I., Broekaert, D., Rosenzweig, A., *et al.* (2022). Altered propionate metabolism contributes to tumour progression and aggressiveness. *Nat Metab*.
- Gomes, A.P., Ilter, D., Low, V., Endress, J.E., Fernandez-Garcia, J., Rosenzweig, A., Schild, T., Broekaert, D., Ahmed, A., Planque, M., *et al.* (2020). Age-induced accumulation of methylmalonic acid promotes tumour progression. *Nature* **585**, 283-287.
- Laurenzana, A., Biagioni, A., Bianchini, F., Peppicelli, S., Chilla, A., Margheri, F., Luciani, C., Pimpinelli, N., Del Rosso, M., Calorini, L., *et al.* (2015). Inhibition of uPAR-TGF β crosstalk blocks MSC-dependent EMT in melanoma cells. *J Mol Med (Berl)* **93**, 783-794.
- Luga, V., Zhang, L., Vitoria-Petit, A.M., Ogunjimi, A.A., Inanlou, M.R., Chiu, E., Buchanan, M., Hosein, A.N., Basik, M., and Wrana, J.L. (2012). Exosomes mediate stromal mobilization of autocrine Wnt-PCP signaling in breast cancer cell migration. *Cell* **151**, 1542-1556.
- Murtha, L.A., Morten, M., Schuliga, M.J., Mabotuwana, N.S., Hardy, S.A., Waters, D.W., Burgess, J.K., Ngo, D.T., Sverdlov, A.L., Knight, D.A., *et al.* (2019). The Role of Pathological Aging in Cardiac and Pulmonary Fibrosis. *Aging Dis* **10**, 419-428.
- Shelke, G.V., Yin, Y., Jang, S.C., Lasser, C., Wennmalm, S., Hoffmann, H.J., Li, L., Gho, Y.S., Nilsson, J.A., and Lotvall, J. (2019). Endosomal signalling via exosome surface TGF β -1. *J Extracell Vesicles* **8**, 1650458.

Wendler, F., Favicchio, R., Simon, T., Alifrangis, C., Stebbing, J., and Giamas, G. (2017). Extracellular vesicles swarm the cancer microenvironment: from tumor-stroma communication to drug intervention. *Oncogene* 36, 877-884.

REVIEWERS' COMMENTS

Reviewer #1 (Remarks to the Author):

The authors have adequately addressed all of my concerns.

I would favor that the data provided under point 4 (SB, Cry, or IL6R suppression) be included in the manuscript.

I agree that the cooperation between IL-6 and veh-EVs is quite interesting, although including these data is not required for full appreciation of the current study.

Reviewer #3 (Remarks to the Author):

I thank the authors for their thoughtful and complete responses to my questions.

Reviewer #1 (Remarks to the Author):

The authors have adequately addressed all of my concerns.

I would favor that the data provided under point 4 (SB, Cry, or IL6R suppression) be included in the manuscript.

I agree that the cooperation between IL-6 and veh-EVs is quite interesting, although including these data is not required for full appreciation of the current study.

Response: As requested, we have included the data provided under point 4 to be included in the manuscript. We thank the reviewer for their time and their contribution to the improvement and finalization of this manuscript.

Reviewer #3 (Remarks to the Author):

I thank the authors for their thoughtful and complete responses to my questions.

Response: We are happy to have addressed your comments satisfactorily and thank the reviewer for their constructive comments in the development of this manuscript.